# Is what's unexpected undesirable? Investigating the expectedness and desirability of gender-stereotypic behaviors in men and women

Hannah B. Eareckson ⓘ*, Madeline E. Heilman

Psychology, New York University, New York, New York, United States of America

* hbe217@nyu.edu

## Abstract

When men and women engage in counter-stereotypic behaviors, they often face penalties for violating gender norms; however, there is evidence that men and women are not always penalized, and are sometimes even rewarded for engaging in counter-stereotypic behaviors. Existing theories of reactions to counter-stereotypic individuals make competing assumptions about the relationship between behavior gender-typicality and desirability: while theories of gender congruity assume that counter-stereotypic behaviors are inherently undesirable, expectancy violation theory assumes that desirability is largely independent of gender-typicality. In six studies, we investigate how gender-typicality affects the expectedness and desirability of twenty gender-stereotypic behaviors. Our results suggests that while behavior expectedness differs according to gender-typicality, judgments of behavior desirability are largely consistent for typical men and women (Studies 1a-1b), Black and Asian men and women (Studies 2a-2b), and successful working men and women (Studies 3a-3b). These results suggest that what is counter-stereotypic is not inherently undesirable and support the application of both expectancy violation theory and the recently proposed unified theory of gender norm violations to predict reactions to gender norm violations.

## Introduction

In research on gender stereotypes and norms, we generally consider agency and communion to encompass masculine and feminine stereotypes, respectively. This is supported by decades of research that suggest agency and communion perceptions are the prime culprits for differences in our perceptions of men and women [1–4]. These stereotypes are thought to be not only descriptive but also prescriptive (gendered "shoulds") and proscriptive (gendered "should nots"), such that individuals think that men should engage in masculine behaviors and women should engage

**Data availability statement:** The data are held in a public repository through OSF - https://osf.io/mpfsy/.

**Funding:** The author(s) received no specific funding for this work.

**Competing interests:** The authors have declared that no competing interests exist.

in feminine behaviors, and that men should not engage in feminine behaviors and women should not engage in masculine behaviors [5,6].

When individuals step out of their prescribed box, they are at risk of both social and economic penalties for counter-stereotypic behaviors [7–9]. For example, agentic and self-promoting women are rated as less likable and less hirable compared to equally qualified, agentic, and self-promoting men [10–12]. There is also evidence that women, but not men, are penalized for engaging in assertive, agentic leadership styles, attempting to exert influence through dominance, or simply succeeding at male-typed tasks [13–16]. While most research on this issue focuses on penalties for women, we see that men, too, are penalized for engaging in feminine-stereotypic behaviors such as self-effacing promotion styles, speaking modestly about past accomplishments, displaying sadness, asking for help, taking family leave, or succeeding in feminized careers such as education [10,17–21].

However, while we frequently see penalties for counter-stereotypic behaviors, these penalties do not always occur—for example, providing evidence of communion can ameliorate penalties for female managers [16], and women who are assigned to leadership roles (rather than seeking them out themselves) are not penalized for their leadership behaviors [22], while women who demonstrate high competence in masculine jobs are sometimes rated more highly than equally qualified men [23]. Additionally, there is growing evidence that women may be rewarded for acting agentically within organizations [24–26]. There is also evidence that men are not unequivocally punished for feminine-stereotypic behaviors: for example, communal men have been rated as equally or more hirable than communal women [27], men who engage in helping behaviors in the workplace are rated more highly than women [28], and men engaging in transformational leadership styles are rated as more promotable than women due to increased communion perceptions [29]. There is even evidence that men who opt to take paternity leave after having children are recommended more highly for rewards and hiring than men who do not, again due to increased communion perceptions [30].

## Predominant explanations for reactions to counter-stereotypic men and women

### Role congruity theories

The lack of fit model [31,32] and role congruity theory [33] posit that penalties for counter-stereotypic behaviors occur because individuals violate prescriptive gender norms. Thus, individuals who engage in counter-stereotypic behaviors are sanctioned for perceived violations of their prescribed social roles [34]. These theories make the assumption that counter-stereotypic behaviors are inherently undesirable and predict that perceivers will penalize women who engage in masculine behaviors, and men who engage in feminine behaviors. They do not, however, account for rewards for counter-stereotypic behaviors.

### Expectancy violation theory

Though it does not specifically deal with gender, expectancy violation theory can help explain the occurrence of rewards as well as penalties for counter-stereotypic

behavior. Key to understanding its relevance is the idea that male and female stereotypes have positive and negative components—something often not attended to in the gender norm violation literature. That is, some counter-stereotypic behavior is positive in valence and other counter-stereotypic behavior is negative in valence (see Table 1). Expectancy violation theory posits that when individuals deviate from expectations, evaluations will become more extreme in the direction of the deviation, producing rewards for positive behavioral deviations and penalties for negative behavioral deviations [35]. Therefore, unlike the lack of fit model and role congruity theory, expectancy violation theory does not assume that what is unexpected is also undesirable; rather, an implicit assumption of expectancy violation theory is that behavior desirability is constant across targets.

Positive masculine stereotypes are intensified prescriptions for men, positive feminine stereotypes are intensified prescriptions for women, negative masculine stereotypes are intensified proscriptions for women, and negative feminine stereotypes are intensified proscriptions for women [5,6].

If behavior desirability were determined entirely by stereotype consistency, then expectancy violation theory would add little to explaining reactions to counter-stereotypical behavior – if behaviors that are counter-stereotypic are inherently undesirable, expectancy violations would induce only penalties, not rewards. But if behavior gender-typicality and desirability are largely independent of one another, expectancy violation theory can help predict both positive and negative reactions to unexpected behavior. It would predict that if an individual acts in a way that is unexpected and desirable, they will be rewarded, and if an individual acts in a way that is unexpected and undesirable, they will be penalized.

## A unified theory of gender norm violations

In Eareckson and Heilman [36], we develop a unified theory of gender norm violations that integrates both role congruity theories and expectation violation theory to explain the observed patterns of reactions to counter-stereotypic men and women. The unified model proposes that the assumptions of expectancy violation theory are generally accurate and can be applied to predict reactions to gender norm violations. However, it adds to the traditional expectancy violation model the moderating effect of gender congruence on desirability, such that individuals may perceive gender-congruent behaviors as more desirable than gender-incongruent behaviors, in line with the predictions of role congruity theories [32,33]. In other words, we argue that there are general social norms that determine which behaviors are desirable or undesirable, but that gender congruity moderates the degree of desirability for these behaviors. More specifically, this theory suggests that although absolute social desirability is largely consistent across gender, there may be differences in relative desirability, such that masculine behaviors may be perceived as relatively more desirable in men than in women, and feminine behaviors may be perceived as relatively more desirable in women than in men.

**Table 1. Positive and Negative, Masculine and Feminine Stereotypes.**

| Valence | Masculine | Feminine |
|---|---|---|
| Positive | • Assertive<br>• Ambitious<br>• Self-reliant<br>• Rational<br>• Decisive<br>• Athletic | • Warm<br>• Friendly<br>• Cooperative<br>• Patient<br>• Sensitive<br>• Cheerful |
| Negative | • Aggressive<br>• Controlling<br>• Stubborn<br>• Dominating<br>• Arrogant<br>• Angry | • Emotional<br>• Yielding<br>• Weak<br>• Naïve<br>• Gullible<br>• Indecisive |

 

## The critical distinction between absolute desirability and relative desirability

We argue that knowledge of the absolute desirability of a behavior – judgments about whether the behavior is desirable or undesirable – is essential for predicting the direction of reactions to counter-stereotypic individuals. If a behavior is judged to be desirable in one gender but undesirable in the other gender, then reactions to men and women exhibiting the same behavior will differ. If, however, as we are proposing, absolute desirability is consistent between men and women, the direction of the reaction to a man or a woman engaging in this behavior will be consistent – i.e., a behavior that is desirable will receive favorable reactions, while a behavior that is undesirable will receive negative reactions regardless of the gender of the individual engaging in the behavior [36]. So absolute desirability is critical to understanding the direction of reactions to counter-stereotypic behavior.

However, the relative desirability of a behavior is not without consequence; a behavior that is absolutely desirable, but nonetheless *less* desirable in one gender than the other, would likely result in a less intense positive reaction to that individual while a behavior that is absolutely undesirable, but nonetheless *more* undesirable in one gender than the other, would likely result in a more intense negative reaction to that individual. Although they should not have an effect on the general direction of the outcome, relative differences in desirability will affect the magnitude of the positive or negative reaction. In other words, relative (but not absolute) differences in behavior desirability may reduce the rewards of a positive expectancy violation, and exacerbate the penalty of a negative expectancy violation, while differences in absolute desirability may result in the same behavior being rewarded in one gender, and penalized in the other. See Fig 1 for a visualization of how differences in behavior desirability would affect reactions to counter-stereotypic behaviors, under the predictions of the unified model.

For differences in expectedness, the distinction between absolute and relative expectedness is not so crucial. According to expectancy violation theory, the degree of expectedness determines how extreme a reaction will be, but not the overall direction of the reaction – that is determined by the absolute desirability, such that a desirable behavior will elicit a positive reaction, while an undesirable behavior will elicit a negative reaction. Assuming constant behavior desirability, it follows that a behavior that is absolutely unexpected will result in the largest reaction, and as the behavior becomes more expected, the magnitude of the reaction will shrink – however, the midpoint is not particularly meaningful; rather, it is the endpoint that is meaningful, when the behavior is absolutely expected; at this point, the unified model would predict no reaction whatsoever, since the behavior is entirely consistent with the existing perception of the individual. A behavior that is completely unexpected will elicit a larger reaction than a behavior that is somewhat unexpected, which will elicit a larger reaction than a behavior that is somewhat expected, which will in turn elicit a larger reaction than a behavior that

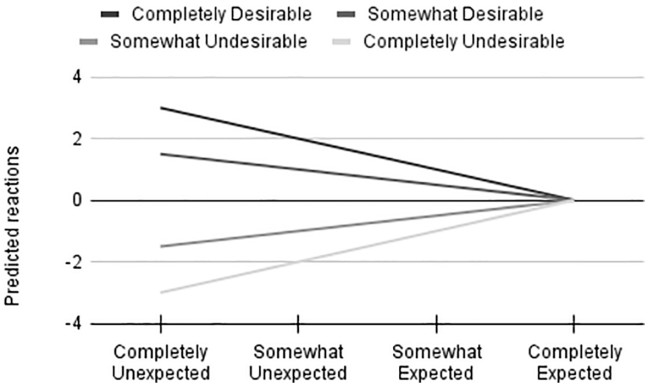

**Fig. 1. Visualization of Predicted Reactions to Behaviors Varying in Expectedness and Desirability.** A reaction above 0 is indicative of a reward (positive change in perception), while a reaction below 0 is indicative of a penalty (negative change in perception).

is completely expected. Regardless of the strength of the reaction, though, it is absolute desirability, not expectedness, that determines whether the reaction is positive or negative. See Fig 1 for a visualization of how expectedness differences should affect reactions to behaviors.

The three theories we have discussed make competing predictions about the relationship between gender-typicality and behavior desirability – for gender congruity theories, gender-typicality should determine absolute desirability; for expectancy violation theory, gender-typicality should have no influence on desirability; and for the unified theory, gender-typicality should affect relative, but not absolute, desirability. Thus, we can use evidence of studies investigating the desirability of gender-typical behaviors in men and women to evaluate the assumptions of these theories.

## Existing evidence of the relationship between gender-typicality and desirability

Past studies investigating perceptions of gender-typicality and desirability have found that what is more typical of one gender than the other is also more desirable in that same gender [5,6,37]. On the face of it, these studies seem to support role congruity theories, such that traits are viewed as more desirable when they are gender-consistent and less desirable when they are gender-inconsistent. However, while these studies are useful and informative for understanding the content of descriptive stereotypes and prescriptive norms for men and women, they are not ideally suited to evaluate the assumptions of these theories. This is because they focus on relative differences in desirability for men and women rather than the absolute desirability differences that are of central concern to us, as well as focus on perceptions of traits rather than behaviors.

While focusing on relative desirability is appropriate to the aim of identifying gender-based prescriptions and proscriptions of differing strength, it can be somewhat misleading when trying to predict positive and negative reactions to counter-stereotypic behavior. For example, Prentice and Carranza [5] label the trait "friendly" as an intensified prescription for women and a relaxed prescription for men, despite the fact that "friendly" was actually rated as highly desirable in both women and men, with ratings of 8.30 and 7.32, respectively, on a scale ranging from 1 to 9. Indeed, "friendly" was rated as more desirable in men than four of the 'intensified prescriptions' for men—consistent (desirability of 7.24 in men), aggressive (desirability of 6.98 in men), intense (desirability of 6.81 in men), and forceful (desirability of 6.58 in men). The reason that consistent, aggressive, intense, and forceful are identified as intensified prescriptions for men, while happy (desirability of 7.49 in men), friendly (desirability of 7.32 in men), helpful (desirability of 7.12 in men), and clean (desirability of 7.05 in men) are not, is because the degree of prescription is identified by relative differences in desirability for men and women, as opposed to absolute desirability ratings. When Eareckson and Heilman [36] re-analyzed these trait data, they found that when you distinguish between relative and absolute desirability, it becomes clear that there is vast overlap between what is absolutely desirable in men, and what is absolutely desirable in women, although there are certain traits for which this does not hold true (e.g., aggressive, forceful, emotional, and yielding).

Additionally, these trait ratings cannot provide direct insight into the perceptions of gender-stereotypic behaviors, as personality traits are inherently more ambiguous in both content and valence than behaviors. For example, a trait like 'assertive' could imply positive behaviors, such as taking charge in a group setting, or negative behaviors, such as interrupting another person when they are speaking. Not only do these traits inherently encompass many types of behaviors, but it also may be that the interpretation of these traits is different for men and women – for example, when one imagines a man being aggressive vs. a woman being aggressive, or a man being emotional vs. a woman being emotional, they may be imagining fundamentally different behaviors, and any similarities or differences observed in these trait perceptions could reflect differences in the underlying behaviors being imagined, rather than differences in perceptions of the *same* behaviors in men and women. To truly test the assumptions of the different theories of reactions to gender norm violations discussed earlier, it is imperative to examine the perceptions of clearly defined behaviors, rather than broad character traits.

In the following research, we address these issues by focusing on absolute behavior desirability in men and women, investigating perceptions of a set of gender-stereotypic behaviors rather than traits, and measuring expectedness directly to more explicitly examine the assumptions of the unified theory of gender norm violations. In addition, there are some methodological limitations that we improve upon in the studies. First, past studies measure typicality, rather than expectedness directly. Though it is probable that typicality and expectedness overlap, it is also possible that stereotype expectedness differs in some way from stereotype typicality; asking about expectedness, rather than typicality, provides the most direct insight into whether or not an expectancy violation exists. Second, many investigations of gender stereotypes and norms aim to avoid social desirability effects by asking participants to evaluate how typical or desirable others in society would view a particular trait in men and women, rather than directly asking participants to provide their own evaluations of stereotypicality or desirability (i.e., Prentice and Carranza (2002) asked participants "how desirable it is *in American society* for a man to possess each of these characteristics"). This is very useful for assessing the perception of societal norms (i.e., how do men and women *think* these behaviors are viewed in society), but not as useful for assessing individuals' own views on these behaviors.

As an example, imagine the case of two participants in a study – Participant A who is actively biased, and thinks society today is too encouraging of women engaging in assertive behaviors, and Participant B who is passionately unbiased, and thinks society unfairly punishes women who engage in assertive behaviors. If we ask our participants to evaluate how others in society view assertiveness in women, Participant A, focusing on the views of others, is likely to indicate assertiveness to be highly desirable, while participant B, focusing on the views of others, is likely to indicate assertiveness to be highly undesirable. These responses would be accurate in reflecting the participants' perceptions of societal norms, but completely reversed if we are trying to understand participants' own views. To understand how individuals will actually react to a woman acting in an assertive manner, they must be asked directly how desirable they, themselves, think these kinds of behaviors are. That is what we do in the current studies, using an online survey format to mitigate social desirability effects as much as possible. Although the motivation to appear unbiased may taint participants' responses – for example, Participant A may not be completely honest in how undesirable they find this behavior – asking participants their direct opinions is the better way to ascertain participants' true views, even if it is imperfect.

## Potential variations in expectedness and desirability

### Intersectionality between race and gender stereotypes

Historically, research investigating perceptions of gender has been conducted with primarily White samples and focused on gender stereotypes and norms regarding White individuals. However, we know that intersectional identities can alter stereotypes, norms, and reactions to counter-stereotypic actors because individuals hold unique and complex stereotypes about individuals of different genders and races [38–40]. There is evidence, for example, that racial stereotypes are gendered, such that stereotypes about Asian individuals are more consistent with feminine stereotypes than masculine stereotypes, while stereotypes about Black individuals are more consistent with masculine stereotypes than feminine stereotypes [41].

The White male norm hypothesis [42] suggests that when race is not explicitly considered, participants are likely to assume and apply social default characteristics to study targets. Thus, when race is not explicitly mentioned in studies with predominantly White samples (the default in nationally representative U.S. samples, as well as for typical Amazon MTurk and Prolific samples), we cannot assume that these results will generalize to perceptions of non-White targets. To understand how the ideas discussed here can be applied to men and women generally, it is therefore critical to study counter-stereotypic behavior not just in 'typical' men and women, who are likely to be perceived as White in prototypical U.S. samples, but also in non-White men and women, for whom behavior expectedness and desirability perceptions may differ.

## Gender stereotypes and norms in the workplace

Although our understanding of gender stereotypes and norms is primarily based on investigations of prototypical men and women in society, many investigations of reactions to counter-stereotypic men and women occur in the workplace setting. However, it is possible that stereotypes and norms for successful working men and women differ from stereotypes and norms for typical men and women. Workplace expectations and requirements in the U.S. are generally centered around men and masculinity norms, and expectations of individuals in workplaces may therefore differ from expectations of individuals in society more broadly—particularly for women, since feminine stereotypes are less aligned with workplace norms than masculine stereotypes [43,44]. To better understand if the theories discussed here can be applied to understanding reactions to counter-stereotypical men and women in the workplace, it is important to examine behavior expectedness and desirability for successful working men and women, as behavior expectedness and desirability for successful working men and women may differ from the expectedness and desirability ratings for typical men and women.

## The current research

In the six studies presented here, we investigate the expectedness and desirability of a set of twenty behaviors for men and women, varying in gender-typicality (masculine and feminine) and valence (positive and negative). We examine perceptions of behavior expectedness and desirability for 'typical' men and women (Studies 1a and 1b), 'typical' Black and Asian men and women (Studies 2a and 2b), and successful working men and women (Studies 3a and 3b). We had three primary goals in pursuing this program of research: (1) to test the assumptions of the unified theory of gender norm violations by examining the expectedness and desirability of a set of gender-stereotypic behaviors in typical men and women; (2) to determine whether our findings generalize to non-White targets by focusing on Black and Asian men and women; and (3) to determine whether our findings generalize to the work context by focusing on successful working men and women.

This work contributes to our theoretical understanding of reactions to gender norm violations by building on and expanding the findings of Eareckson and Heilman (2024). In their paper, they re-analyze trait data collected by Prentice and Carranza (2002) and Rudman et al. (2012) and find support for the unified model – specifically, they find that absolute desirability of traits is largely consistent for typical men and women, while relative desirability differs. However, the trait data they re-analyze suffers from the key limitations discussed earlier: traits are ambiguous, and these perceptions may not generalize to perceptions of behaviors; the ratings of 'expectedness' were actually ratings of 'typicality' of men and women in society; the ratings exclusively examine perceptions of prototypical men and women; and due to the way the questions were asked, the ratings may reflect perceptions of societal norms, rather than individuals' own views. By systematically testing the assumptions of the unified model with behaviors, rather than traits, and by examining the perceptions of these behaviors in not only typical men and women, but also non-White and working men and women, we build on the findings of Eareckson and Heilman (2024) and provide more direct insight into how specific gender-stereotypic behaviors are perceived in the US today.

**Hypothesis 1: Behaviors will differ in relative expectedness by target gender, such that behaviors are rated as more expected in the gender they are stereotypic in.** If we see a significant interaction between target gender and behavior gender-typicality in Studies 1a, 2a, and 3a, such that feminine-stereotypic behaviors are more expected of women than men and masculine-stereotypic behaviors are more expected of men than women, this provides evidence that behaviors are relatively more expected in the gender they are stereotypic of. In Study 2a, we also test for a two-way interaction between target race and behavior stereotypicality, as well as a three-way interaction between target race, target gender, and behavior stereotypicality, to examine whether or not these gender stereotypes are racialized.

**Hypothesis 2: Behaviors will be consistent in absolute desirability, regardless of gender-typicality.** If positive behaviors are rated as desirable in both men and women (95% CIs above 0) and negative behaviors are undesirable in both men and women (95% CIs below 0) in Studies 1b, 2b, and 3b, these results will support our hypothesis.

**Hypothesis 3: Behaviors will differ in relative desirability by target gender, such that behaviors are rated as more desirable in the gender they are stereotypic in.** If we see a significant interaction between target gender and behavior gender-typicality in Studies 1b, 2b, and 3b, such that feminine-stereotypic behaviors are more desirable in women than men and masculine-stereotypic behaviors are more desirable in men than women, this provides evidence that behaviors are relatively more expected in the gender they are stereotypic of. This finding would be consistent with past research on stereotypes and norms for men and women [5,6,37]. In Study 2b, we also test for a two-way interaction between target race and behavior stereotypicality, as well as a three-way interaction between target race, target gender, and behavior stereotypicality, to examine whether or these gender prescriptions are racialized.

## Materials and methods

### Ethics statement

NYU Washington Square (WSQ) IRB approved these studies (IRB-FY2016–479); written consent was obtained.

### Material development and piloting

In developing the set of 20 behaviors, we utilized the trait stereotypes and norms identified by past research on descriptive gender stereotypes and prescriptive norms [5,6] as the foundation for generating five masculine positive (e.g., takes charge in group settings), five masculine negative (e.g., brags about their accomplishments), five feminine positive (e.g., goes out of their way to help others), and five feminine negative (e.g., always backs down from confrontations) behaviors. We pilot tested these behaviors with 100 Amazon MTurk raters. Participants were asked to classify each of the twenty behaviors as stereotypically masculine or stereotypically feminine, and as positive or negative; we tested each behavior categorization with chi square goodness of fit tests comparing the observed distributions to the 50/50 distributions that would be consistent with chance.

This pilot test confirmed that participant perceptions of all twenty behaviors were consistent with our theoretical classifications, with all ten masculine behaviors classified as significantly more masculine than feminine (min = interrupts others at 72% masculine, max = plays to win at 92% masculine), all ten feminine behaviors classified as significantly more feminine than masculine (min = maintains positive attitude at 61% feminine, max = cries easily at 91% feminine), all ten positive behaviors classified as significantly more positive than negative (min = plays to win at 83% positive, max = maintains positive attitude at 97% positive), and all ten negative behaviors classified as significantly more negative than positive (min = backs down from confrontations at 71% negative, max = interrupts others at 90% negative). These results indicate that all twenty behaviors are effective manipulations of gender stereotypicality and valence in the intended direction.

### Procedure

Participants consented to the survey and were asked to answer the survey questions honestly and pay attention to each behavior they were asked to respond to. They then were presented with twenty stimulus behaviors and instructed to rate the expectedness or desirability of each behavior for their assigned target by answering the question:

"How [expected/desirable] do you think this behavior is for a [TARGET]?" The target in Studies 1a and 1b was a typical man or woman, the target in Studies 2a and 2b was a typical Black man, Black woman, Asian man, or Asian man, and the target in Studies 3a and 3b was a successful working man or woman.

Participants rated each of the twenty behaviors on a seven-point scale from 'extremely [unexpected/undesirable]' to 'extremely [expected/desirable]'. Finally, participants provided demographic information about their gender identity, race identity, age, and political orientation. Participants were randomly assigned to one of the possible targets in each study and evaluated all twenty behaviors in Table 2 in randomized order. Data collection for these studies began on 02/18/2021 and was completed on 04/16/2022.

**Table 2. Behaviors Investigated in Studies 1a-3b.**

| Category | Behavior |
|---|---|
| Masculine Positive | • Takes charge in group settings<br>• Negotiates for the best deal on products and services [*Negotiates for higher raises and bonuses*]<br>• Speaks positively about their accomplishments<br>• Seeks leadership roles [*Seeks opportunities for advancement*]<br>• Plays to win in games and sports [*Competes with co-workers for high profile assignments*] |
| Masculine Negative | • Often interrupts or speaks over others when they are talking<br>• Never admits they are wrong<br>• Brags about their accomplishments<br>• Loses their temper when frustrated<br>• Never asks for or accepts help on projects |
| Feminine Positive | • Goes out of their way to help others<br>• Anticipates and takes care of others' needs<br>• Maintains a positive attitude<br>• Takes care of household chores [*Offers to take notes in meetings*]<br>• Plans parties for friends and family [*Plans parties for co-workers' birthdays*] |
| Feminine Negative | • Apologizes often, even when they have done nothing wrong<br>• Always backs down from confrontations<br>• Cries easily when frustrated<br>• Is sensitive and their feelings are easily hurt<br>• Lets others interrupt them and speak over them |

Behavior modifications for Studies 3a-3b to be consistent with the successful working role are indicated in brackets.

## Analyses

To test hypotheses 1 and 3, we fit linear mixed models for the effect of the interaction between target gender (effects coded, men = −1 and women = 1) and behavior gender-typicality (effects coded, masculine = −1 and feminine = 1) on expectedness and desirability, respectively, including random intercepts for participant ID and specific behavior. We check if participant gender, participant race, age, and political orientation moderate these interactions across all studies, by including them as interaction terms in the linear mixed models. Participant gender was effects coded, male = −1, female = 1, other = 0; participant race was broadly construed and dummy coded, White = 1, Non-White = 0, age was measured with a numeric text entry, and political orientation was measured on a 7-point scale from strongly conservative (−3) and strongly liberal (3). In Studies 2a and 2b, we added target race (effects coded, Asian = −1 and Black = 1) to the model, to test for both the two-way interaction between target race and behavior gender-typicality, as well as the three-way interaction between target gender, target race, and behavior-typicality. We also tested whether participant gender and race moderate these interactions in Studies 2a-2b. To test hypothesis 2, we analyzed the 95% confidence intervals on the behavior-level means to identify whether each positive behavior was viewed as desirable for both men and women, and whether each negative behavior was viewed as undesirable for both men and women. Data were analyzed using RStudio 2023.09.1 for Windows. All data and analyses are available at https://osf.io/mpfsy/?view_only=89be74a03d9a4decacb03c5143b62a5d.

## Participants

In all studies, Amazon MTurk workers were recruited from CloudResearch and were compensated $0.40 to participate in the study online. In Studies 1a, 1b, 3a, and 3b, no demographic quotas were set, and sample demographics represent the general MTurk population that elected to take part in our studies. In Studies 2a and 2b, we set quotas for our sample to recruit 50% White participants and 50% non-White participants; since we were examining perceptions of behaviors in

non-White targets, we wanted to ensure that our sample was not overwhelmingly White. See Table 3 for an overview of each study's question wording, hypotheses, and participant demographic summary.

## Results

### Study 1a: Expectedness in typical targets

**Hypothesis 1: Relative differences in expectedness by gender.**  As predicted in Hypothesis 1, there was a significant two-way interaction between target gender and behavior gender-typicality, $b = .18$, $t(3666) = 8.48$, $p < .001$, such that feminine-stereotypic behaviors were more expected of women ($M = 4.16$, $SD = 1.35$) than men ($M = 2.22$, $SD = 1.51$), and masculine-stereotypic behaviors were more expected of men ($M = 4.18$, $SD = 1.25$) than women ($M = 3.04$, $SD = 1.54$). See Fig 2 for a visualization of this interaction.

**Tests of moderations by participant identity.**  Participant gender moderated this interaction, $b = .20$, $t(1878) = 6.65$, $p < .001$, such that the interaction was significant and consistent in direction for both male and female participants, but stronger for female participants, $b = .94$, $t(1082) = 24.85$, $p < .001$, than for male participants, $b = .55$, $t(740) = 11.98$, $p < .001$. Participant race also moderated this interaction, $b = .30$, $t(1878) = 4.54$, $p < .001$, such that the interaction was significant and consistent in direction for both White and Non-White participants, but stronger for White participants, $b = .86$, $t(1329) = 24.99$, than for Non-White participants, $b = .57$, $t(531) = 10.22$, $p < .001$. Participant political orientation moderated the interaction between target gender and behavioral stereotypicality, $b = .09$, $t(1830) = 4.74$, $p < .001$, such that it was stronger for liberal participants than conservative participants. Participant age was not a significant moderator, $b < .01$, $t(1840) = .60$, $p = .549$.

**Variation in effect sizes across behaviors.**   We saw large effect sizes for each item, ranging from $d = .46$ for losing one's temper to $d = 2.73$ for crying when frustrated; the average effect size across all twenty behaviors was $d = 1.27$ ($SD = .66$). See Fig 3 for the behavior-level expectedness ratings.

**Summary.**   Behaviors were more expected in the gender they are stereotypic of (hypothesis 1 supported). We saw large effect sizes for the difference in expectedness for all items studied, suggesting that gender has a very large influence on the expectedness of the twenty behaviors studied here for typical men and women.

### Study 1b: Desirability in typical targets.

**Hypothesis 2: Absolute consistency in desirability.**   For both male and female targets, we saw consistent absolute desirability ratings across all twenty behaviors studied – all five positive masculine-stereotypic behaviors and all five positive feminine-stereotypic behaviors were seen as desirable in both men and women (95% CIs above 0), while all five negative masculine-stereotypic behaviors and all five negative feminine-stereotypic behaviors are seen as undesirable in both men and women (95% CIs below 0, see Fig 4 for a visualization).

**Hypothesis 3: Relative differences in desirability by gender.**   Surprisingly, we did not find an interaction between target gender and behavior gender-typicality, $b < .01$, $t(2827) = 1.24$, $p = .216$, indicating that relative behavior desirability did not differ according to behavioral gender consistency (see Fig 5).

**Tests of moderations by participant identity.**  Participant gender did not moderate this interaction, $b = -.24$, $t(1878) = .84$, $p = .404$, nor did participant race, $b < -.01$, $t(1878) = .03$, $p = .974$, nor age, $b < .01$, $t(1859) = 1.76$, $p = .079$. Participant political orientation moderated the interaction between target gender and behavior stereotypicality, $b = -.04$, $t(1859) = 2.64$, $p = .008$, such that more conservative participants showed the predicted interaction, rating feminine behaviors as relatively more desirable in women than in men, and masculine behaviors as relatively more desirable in men than women, but more liberal participants did not.

**Variation in effect sizes across behaviors.**   Effect sizes for individual behaviors varied widely, from $d = .01$ for interrupting others to $d = .87$ for crying when frustrated; the average effect size across all twenty behaviors was $d = .25$ ($SD = .21$).

**Table 3. Overview of studies and participant demographics.**

| Study | Question and Hypotheses Tested | Hypotheses tested | Participant Demographics |
|---|---|---|---|
| 1a N = 100 | How expected do you think this behavior is for a TYPICAL [WOMAN/MAN]?" | **Hypothesis 1:** Differences in relative expectedness by gender | 100 participants included; 3 excluded 40% male, 58% female, 2% nonbinary or not identified 71% White, 11% Asian, 8% Hispanic, 6% Black, 4% other or not identified Median age 32, ranging from 18–68 Moderate-liberal political orientation, $M = .33$, $SD = 1.57$ |
| 1b N = 100 | How desirable do you think this behavior is for a TYPICAL [WOMAN/MAN]?" | **Hypothesis 2:** Consistency in absolute desirability by gender **Hypothesis 3:** Differences in relative desirability by gender | 100 participants included; 2 excluded 51% male, 47% female, 2% nonbinary or not identified 77% White, 7% Black, 6% Asian, 5% Hispanic, 5% other or not identified Median age 32, ranging from 20–73 Moderate-liberal political identity ($M = .47$, $SD = 1.76$) |
| 2a N = 201 | How expected do you think this behavior is for a TYPICAL [BLACK/ASIAN] [WOMAN/MAN]?" | **Hypothesis 1:** Differences in relative expectedness by gender and race | 201 participants included, 3 excluded 35% male, 63% female, 2% nonbinary or not identified 50% White, 20% Black, 14% Hispanic, 9% Asian, 7% other or not identified Median age 34, ranging from 18–72 Moderate-liberal political identity ($M = .49$, $SD = 1.66$) |
| 2b N = 199 | How desirable do you think this behavior is for a TYPICAL [BLACK/ASIAN] [WOMAN/MAN]?" | **Hypothesis 2:** Consistency in absolute desirability by gender **Hypothesis 3:** Differences in relative desirability by gender and race | 199 participants included, 4 excluded 39% male, 59% female, 2% nonbinary or not identified 53% White, 20% Black, 12% Hispanic, 11% Asian, 4% other or not identified Median age 33, ranging from 18–70 Moderate-liberal political identity ($M = .64$, $SD = 1.59$) |
| 3a N = 194 | How expected do you think this behavior is for a SUCCESSFUL WORKING [WOMAN/MAN]?" | **Hypothesis 1:** Differences in relative expectedness by gender | 194 participants included, 3 excluded 44% male, 55% female, 1% nonbinary or not identified 72% White, 10% Hispanic, 10% Asian, 5% Black, 3% other or not identified Median age 31, ranging from 18–71 |
| 3b N = 197 | How desirable do you think this behavior is for a SUCCESSFUL WORKING [WOMAN/MAN]?" | **Hypothesis 2:** Consistency in absolute desirability by gender **Hypothesis 3:** Differences in relative desirability by gender | 107 participants included, 2 excluded 40% male, 57% female, 3% nonbinary or not identified Moderate-liberal political identity ($M = .43$, $SD = 1.74$) 76% White, 9% Black, 7% Asian, 4% Hispanic, 4% other or not identified Median age 31, ranging from 18–78 Moderate-liberal political identity ($M = .73$, $SD = 1.64$) |

**Summary.** Consistent with our predictions, we found that positive behaviors were rated as absolutely desirable in both men and women and negative behaviors were rated as absolutely undesirable in both men and women, regardless of gender stereotypicality (hypothesis 2 supported). Inconsistent with our predictions, there was no interaction between target gender and behavior gender-typicality on the desirability ratings, suggesting that not only was absolute desirability consistent between men and women, but relative desirability was consistent as well (hypothesis 3 not supported).

## Study 2a: Expectedness in Black and Asian targets

**Hypothesis 1: Relative differences in expectedness by gender.** As predicted, there was a significant interaction between target gender and behavior gender-typicality, $b = .17$, $t(3797) = 7.75$, $p < .001$, such that masculine-stereotypic behaviors were more expected of men ($M = 3.40$, $SD = 1.50$) than women ($M = 3.27$, $SD = 1.60$) and feminine-stereotypic behaviors were more expected of women ($M = 3.43$, $SD = 1.60$) than men ($M = 2.87$, $SD = 1.59$).

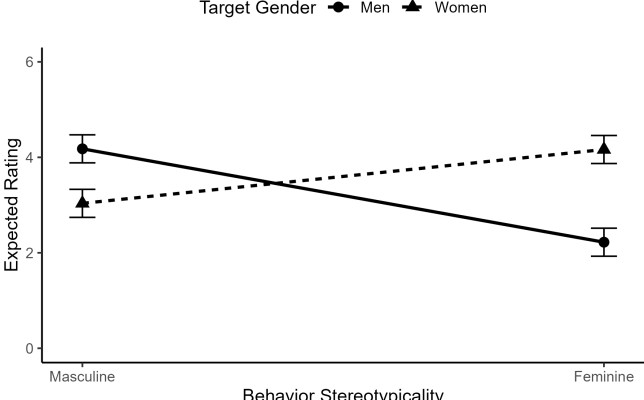

**Fig 2. The interaction between behavior gender-typicality and target gender in ratings of behavior expectedness for typical men and women.**

We also saw a significant three-way interaction between target gender, target race, and behavior gender-typicality, $b = -.06$, $t(3797) = 2.73$, $p = .006$. When we decompose this by looking at the two-way interaction between target gender and behavior gender-typicality for both Asian targets, $b = .23$, $t(1861) = 7.08$, $p < .001$, and Black targets, $b = .11$, $t(1918) = 3.90$, $p < .001$, we see that the two-way interaction between target gender and behavior gender-typicality is significant for both Asian and Black men and women; however, the interaction is larger for Asian targets, relative to Black targets. This seems to be driven by a higher expectation of masculine behaviors in Asian men ($M = 3.30$, $SD = 1.56$) than in Asian women ($M = 2.91$, $SD = 1.73$), but similar expectations of masculine behaviors in Black men ($M = 3.50$, $SD = 1.33$) and in Black women ($M = 3.64$, $SD = 1.37$). See Fig 6 for a visualization.

### Tests of moderations by participant identity.

Participant gender did not moderate the two-way interaction between target gender and behavior stereotypicality, $b = .03$, $t(3793) = 1.27$, $p = .204$, nor did participant race, $b = .01$, $t(3793) = .21$, $p = .836$, age, $b < .01$, $t(3774) = 1.08$, $p = .282$, or political orientation, $b = -.01$, $t(3774) = .80$, $p = .424$.

### Variation in effect sizes across behaviors

We observed variable effect sizes across behaviors for the expectedness differences between Asian men and women, ranging from $d = .02$ (helps others) to $d = 1.22$ (does chores), with an average effect size of $d = .35$ ($SD = .29$). For Black men and women, we observed effect sizes ranging from $d = .06$ (lets others interrupt) to $d = 1.06$ (does chores), with an average effect size of $d = .38$ ($SD = .28$). See Fig 7 for the behavior-level expectedness ratings.

**Relative differences in expectedness by race.** There was a significant interaction between target race and behavior gender-typicality, $b = -.20$, $t(3797) = 9.14$, $p < .001$, such that masculine-stereotypic behaviors were more expected of typical Black individuals ($M = 3.56$, $SD = 1.35$) than of typical Asian individuals ($M = 3.09$, $SD = 1.66$), and feminine-stereotypic behaviors were more expected of typical Asian individuals ($M = 3.33$, $SD = 1.59$) than typical Black individuals ($M = 2.99$, $SD = 1.63$).

After decomposing the significant three-way interaction, the two-way interaction between target race and behavior gender-typicality remained significant for both men, $b = -.14$, $t(1861) = 4.57$, $p < .001$, and women, $b = -.26$, $t(1918) = -8.41$, $p < .001$, though it was stronger for women than for men.

**Summary.** Behaviors were relatively more expected in the gender they are stereotypic of (hypothesis 1 supported); however, a significant three-way interaction between target gender, target race, and behavior gender-typicality suggests

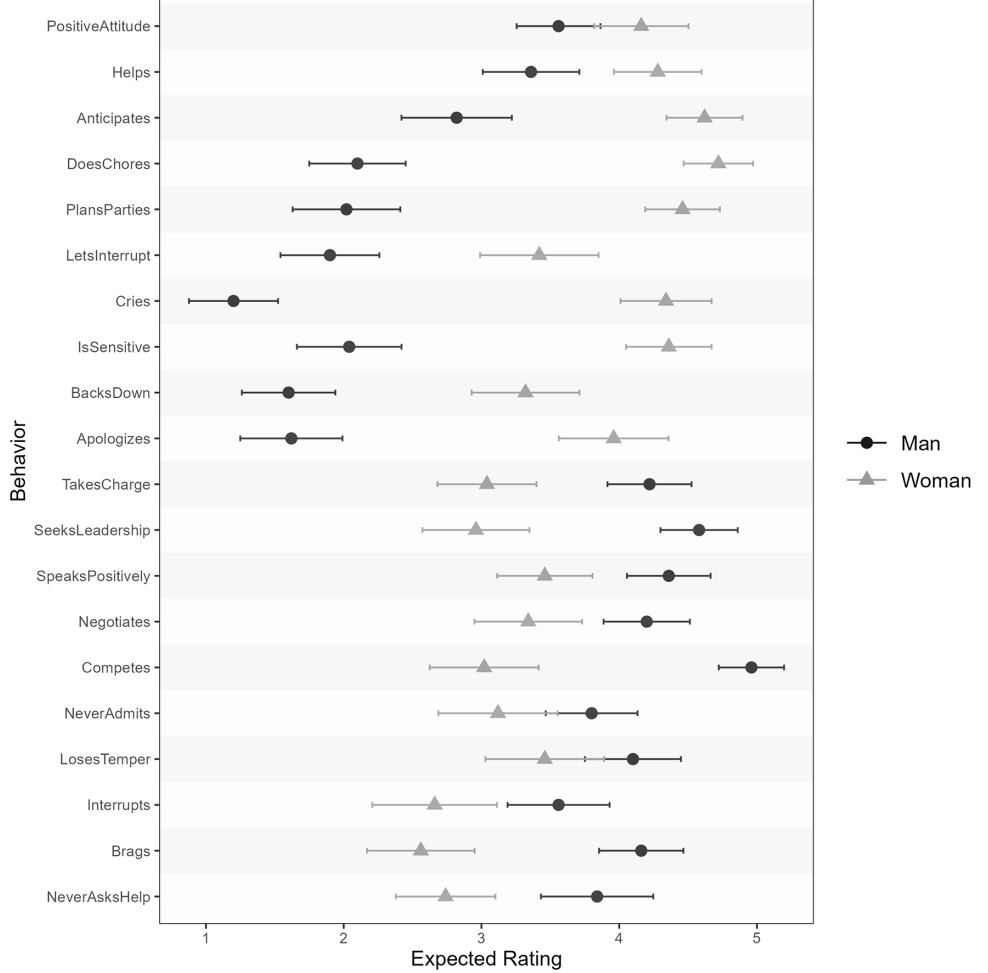

**Fig 3. Behavior-level expectedness means with 95% CIs for typical men and women.**

that this effect is stronger for Asian men and women than for Black men and women. In particular, it is notable that the condition means for the expectedness of masculine behaviors for Black men and women were extremely similar, suggesting that the expectedness differences we observe for masculine behaviors hold for prototypical men and women and for Asian men and women, but may not generalize to Black men and women. Additionally, we saw smaller effect sizes on the behavior-level differences in expectedness for Asian and Black men and women, compared to in Study 1a. Altogether, these results suggest that our conclusions from Study 1a partially generalize to typical Asian and Black men and women, although information about racial identity may weaken the effect of target gender on behavioral expectations, since we saw smaller effect sizes of gender on behavior expectedness in this study, compared to in Study 1a.

In addition to the effects of target gender, we found evidence for an interaction between target race and gender on behavior expectedness, such that feminine behaviors were more expected of Asian targets than of Black targets, and masculine behaviors were more expected of Black targets than of Asian targets. This is consistent with past work suggesting that race is gendered, such that Asian individuals are stereotyped as feminine, and Black individuals are stereotyped as masculine [45,46]. Finally, we found evidence that participant race moderated the three-way interaction between target gender, target race, and behavior stereotypicality; specifically, White participants applied gender stereotypes to both Black

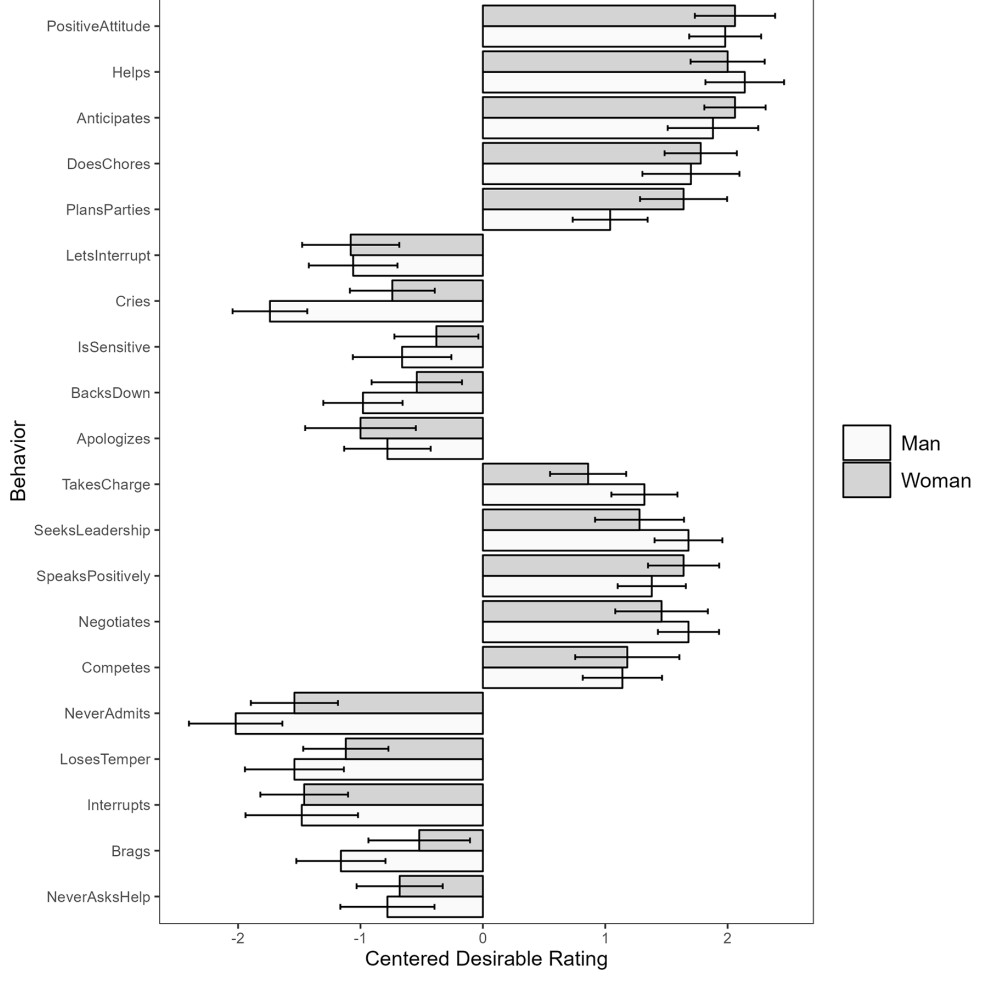

**Fig 4. Behavior-level desirability means with 95% CIs for typical men and women.**

and Asian targets, while non-White participants applied gender stereotypes only to Asian targets. Further exploring this effect in Black and Asian participants, specifically, we find that this effect generally holds true for both Black and Asian participants, with Black participants only marginally applying gender stereotypes to Black targets, and Asian participants applying no significant gender stereotypes to Black targets. This moderation of gender stereotyping by participant race is very interesting and suggests that across race identities, participants stereotype Asian targets according to gender, but that only White participants seem to significantly stereotype Black targets according to gender (Black participants applied marginal, but nonsignificant, gender stereotyping to Black targets).

### Study 2b: Desirability in Black and Asian targets

**Hypothesis 2: Absolute consistency in desirability.** We saw consistent absolute desirability for Asian men and women across nineteen of the twenty behaviors, with all five positive feminine behaviors and all five positive masculine behaviors seen as absolutely desirable for both Asian men and Asian women (95% CIs above 0), and all five negative feminine behaviors seen as absolutely undesirable for both Asian men and Asian women (95% CIs below 0). We saw four out of five negative masculine behaviors rated as absolutely undesirable for both Asian men

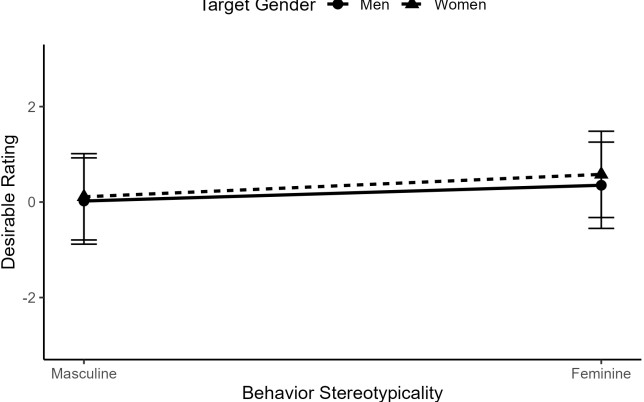

**Fig 5. The interaction between behavior gender-typicality and target gender in ratings of behavior desirability for typical men and women.**

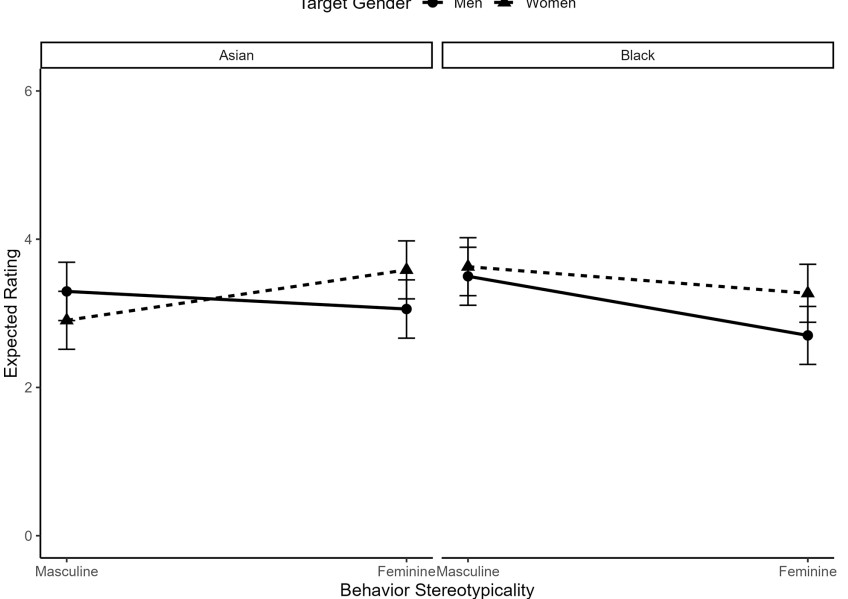

**Fig 6. The interaction between behavior gender-typicality and target gender in ratings of behavior expectedness for typical Asian and Black men and women.**

and Asian women (95% CIs below 0); the only exception was "never asking for help," which was rated as absolutely undesirable for Asian women (95% CI [−.78 −.17]), but neither absolutely desirable nor undesirable for Asian men (95% CI [-.65.17]).

We observed consistent absolute desirability for all twenty behaviors for Black men and women, with all five positive feminine behaviors and all five positive masculine behaviors seen as absolutely desirable for both Black men and Black women (95% CIs above 0), and all five negative feminine behaviors seen as absolutely undesirable for both Black men and black women (95% CIs below 0). Again, we saw four out of five negative masculine behaviors rated as absolutely undesirable for both Black men and Black women (95% CIs below 0); the only exception was "brags

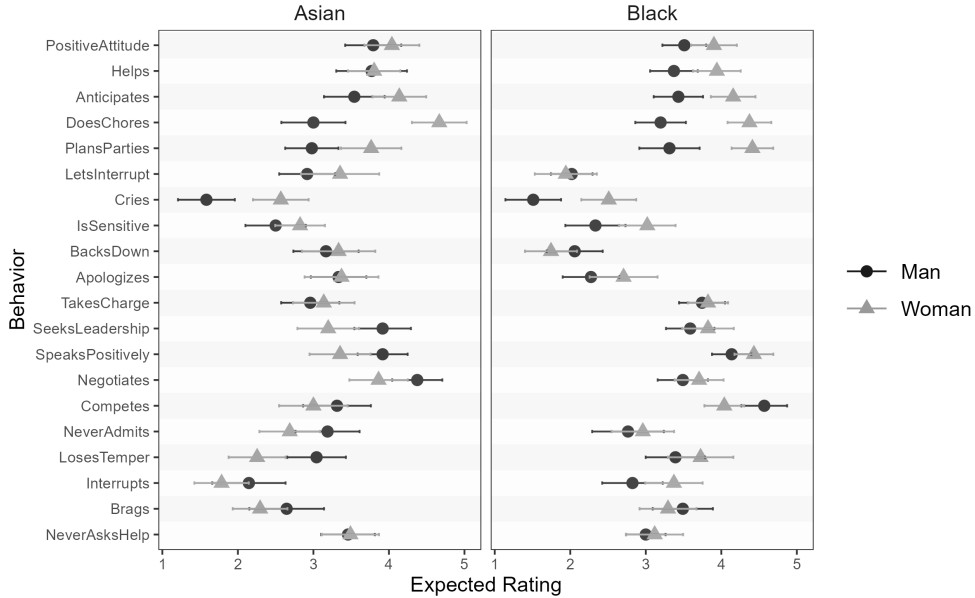

**Fig 7. Behavior-level expectedness means with 95% CIs for typical Asian and Black men and women.**

about accomplishments," with the 95% CI including 0 for both Black men (95% CI [-.92.04]) and Black women (95% CI [-.69.14]). See Fig 8 for a graph of the behavior-level means.

**Hypothesis 3: Relative differences in desirability by target gender.** We found a significant interaction between target gender and behavior gender-typicality, $b=.05$, $t(3759) = 2.30$, $p=.021$, indicating that masculine-stereotypic behaviors were relatively more desirable in men ($M=.13$, $SD=1.96$) than in women ($M=.08$, $SD=1.89$), while feminine-stereotypic behaviors were relatively more desirable in women ($M=.42$, $SD=1.87$) than in men ($M=.29$, $SD=1.94$).

However, this was qualified by a significant three-way interaction between target race, target gender, and behavior gender-typicality, $b=-.08$, $t(3759) = -3.61$, $p<.001$, such that the two-way interaction between target gender and behavior gender-typicality was significant for Asian targets, $b=.13$, $t(1842) = 4.41$, $p<.001$, but not Black targets, $b=-.03$, $t(1899) = -.89$, $p=.375$. This suggests that feminine behaviors were relatively more desirable in Asian women ($M=.61$, $SD=1.76$) than Asian men ($M=.28$, $SD=181$), and masculine behaviors were relatively more desirable in Asian men ($M=.07$, $SD=1.87$) than Asian women ($M=-.12$, $SD=1.81$), but that we saw similar desirability of feminine behaviors for Black women ($M=.24$, $SD=1.94$) and Black men ($M=.19$, $SD=2.06$), as well as of masculine behaviors for Black men ($M=.29$, $SD=2.06$) and Black women ($M=.26$, $SD=1.94$). See Fig 9 for a visualization of these interactions.

### Tests of moderations by participant identity.

Participant gender did not moderate the two-way interaction between target gender and behavior gender-typicality, $b<.01$, $t(3755) =.20$, $p=.838$, nor did participant race, $b<.01$, $t(3755) =.06$, $p=.950$, nor political orientation, $b=-.01$, $t(3736) = 1.16$, $p=.247$. However, the interaction was moderated by age, $b<.01$, $t(3755) = 3.37$, $p<.001$, such that the two-way interaction between target gender and behavior typicality was stronger for older, than for younger, participants.

### Variation in effect sizes across behaviors.

We observed variable effect sizes across behaviors for the desirability difference between Asian men and women, ranging from $d<.01$ (apologizes often) to $d=.62$ (cries when frustrated), with an average effect size of $d=.26$ ($SD = .17$). For

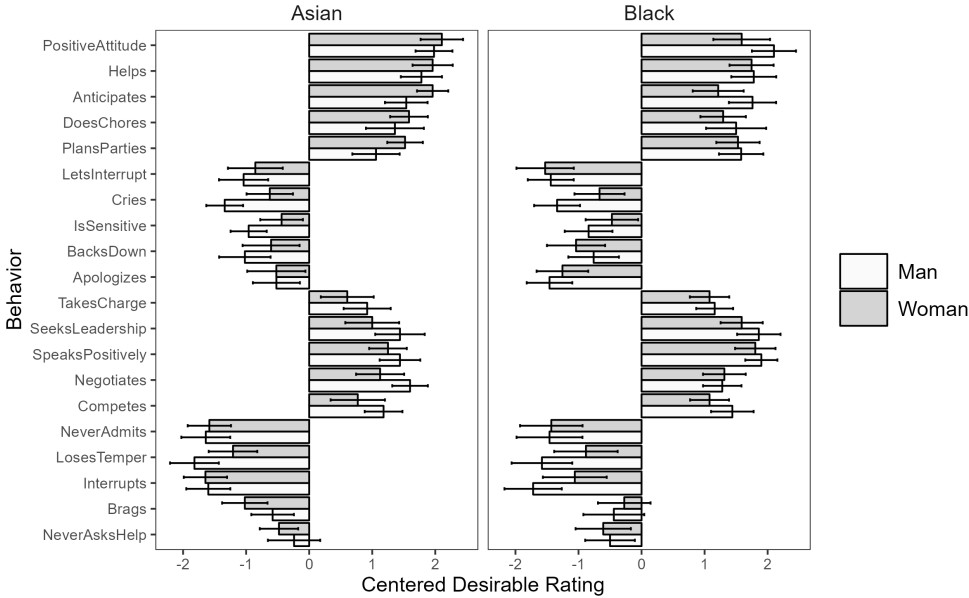

**Fig 8. Behavior-level desirability means for typical Asian and Black men and women.**

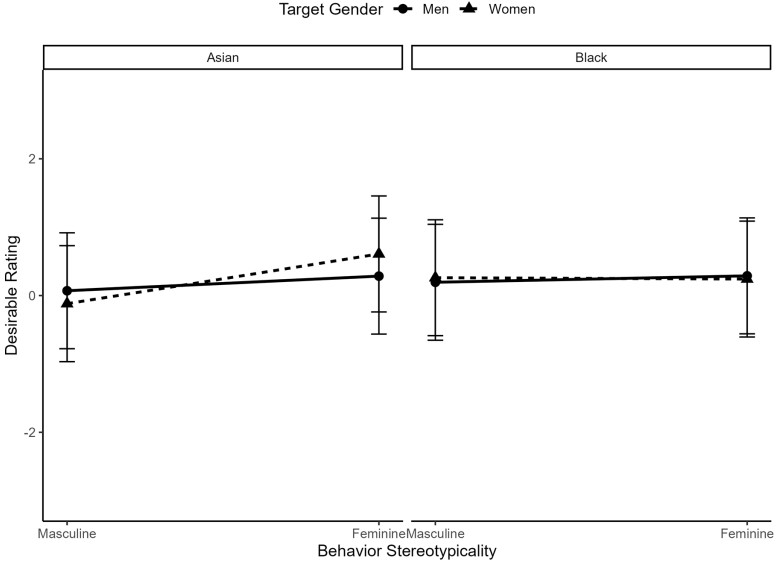

**Fig. 9. The interaction between behavior gender-typicality and target gender in ratings of behavior desirability for typical Asian and Black men and women.**

Black men and women, we observed effect sizes ranging from $d = .02$ (never admits when they are wrong) to $d = .50$ (cries when frustrated), with an average effect size of $d = .19$ ($SD = .15$).

**Relative differences in desirability by target race.** We found a significant interaction between target race and behavior gender-typicality, $b = -.11$, t(3759) = 4.99, $p < .001$, such that masculine behaviors were rated as relatively more

desirable in Black ($M = .23$, $SD = 1.98$) than Asian ($M = -.02$, $SD = 1.84$) targets, while feminine behaviors were rated as relatively more desirable in Asian ($M = .44$, $SD = 1.80$) than Black ($M = .26$, $SD = 2.00$) targets.

When decomposing the three-way interaction between target race, target gender, and behavior gender-typicality, we found that the interaction between target race and behavior gender-typicality was significant only for women, $b = -.19$, $t(1861) = 5.99$, $p < .001$, and not for men, $b = -.03$, $t(1880) = 1.00$, $p = .318$. This suggests that in particular, we see that feminine behaviors were more relatively desirable in Asian women ($M = .61$, $SD = 1.76$) than in Black women ($M = .24$, $SD = 1.94$), and masculine behaviors were relatively more desirable in Black women ($M = .26$, $SD = 1.94$) than in Asian women ($M = -.12$, $SD = 1.81$), but that we saw similar desirability of feminine behaviors for Black men ($M = .19$, $SD = 2.06$) and Asian men ($M = .28$, $SD = 1.81$), and of masculine behaviors for Black men ($M = .29$, $SD = 2.06$) and Asian men ($M = .07$, $SD = 1.87$).

**Summary.** Consistent with Study 1b, we found that absolute behavior desirability was largely consistent for both Black and Asian men and women; the only exception was "never asks for or accepts help on projects," which was absolutely undesirable in Asian men, but neither undesirable nor desirable in Asian women (hypothesis 1 largely supported). Additionally, we saw that for both Black men and women, the behavior "brags about their accomplishments" was neither undesirable nor desirable, as the confidence interval included zero. Unlike in Study 1b, we found an interaction between target gender and behavior gender-typicality on desirability, such that behaviors were rated as more desirable in the gender they are stereotypic of; however, this was qualified by a three-way interaction, indicating that the interaction was driven by differences in behavior desirability for Asian men and women, but not for Black men and women (hypothesis 3 partially supported). Overall, the result patterns for absolute desirability are consistent with those of Study 1b, indicating that the absolute desirability of behaviors is largely consistent for Asian men and women, as well as Black men and women; in addition, these results also provide some support for differences in relative desirability according to gender, particularly for Asian men and women.

### Study 3a: Expectedness in successful working targets

**Hypothesis 1: Relative differences in expectedness.** As predicted, there was a significant two-way interaction between target gender and behavior gender-typicality, $b = .18$, $t(3666) = 8.48$, $p < .001$, such that feminine-stereotypic behaviors were relatively more expected of successful working women ($M = 3.73$, $SD = 1.94$) than successful working men ($M = 3.50$, $SD = 1.96$), and masculine-stereotypic behaviors were relatively more expected of successful working men ($M = 4.84$, $SD = 1.89$) than successful working women ($M = 4.32$, $SD = 2.00$). See Fig 10 for a visualization of this interaction.

### Tests of moderations by participant identity.

Participant gender did not moderate the interaction between target gender and behavior typicality, $b = .03$, $t(3664) = 1.45$, $p = .147$; nor did participant race, $b = -.04$, $t(3664) = .78$, $p = .437$. Participant age did moderate this interaction, $b < -.01$, $t(3645) = 4.20$, $p < .001$, such that the interaction was stronger for younger participants than for older participants. Participant political ideology also moderated the interaction, $b = .07$, $t(3664) = 5.20$, $p < .001$, such that the interaction was stronger for more liberal participants than for more conservative participants.

**Variation in effect sizes across behaviors.** Effect sizes ranged from $d < .01$ (speaks positively about accomplishments) to $d = .64$ (brags about accomplishments); the mean effect size across behaviors was $d = .29$ ($SD = .21$). See Fig 11 for a graph of the behavior-level means.

**Summary.** As in Studies 1a and 2a, there was a significant two-way interaction between target gender and behavior gender-typicality in the predicted direction, with behaviors rated as more expected in the gender they are stereotypic of (hypothesis 1 supported). However, the effect sizes on the differences in behavior desirability for men and women were

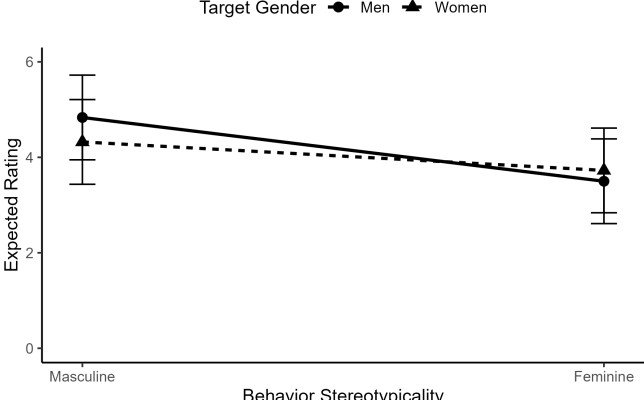

**Fig 10. The interaction between behavior gender-typicality and target gender in ratings of behavior expectedness for successful working men and women.**

much smaller for successful working targets, compared to typical men and women; the effect sizes in this study were even smaller than the effect sizes observed in Study 2b for typical Black and Asian men and women. This suggests that information indicating that an individual is successful in the workplace may result in smaller effects of gender on the expectedness of stereotypically masculine and feminine behaviors, compared to the effects of gender on expectedness in typical targets.

### Study 3b: Desirability in successful working targets

**Hypothesis 2: Absolute consistency in desirability.** For both male and female targets, we saw consistent absolute desirability ratings across all twenty behaviors studied – all five positive masculine-stereotypic behaviors and all five positive feminine-stereotypic behaviors are seen as desirable in both men and women (95% CIs above 0), while all five negative masculine-stereotypic behaviors and all five negative feminine-stereotypic behaviors are seen as undesirable in both men and women (95% CIs below 0). See Fig 12 for a visualization of the behavior-level means.

**Hypothesis 3: Relative differences in desirability.** We found a significant two-way interaction between target gender and behavior gender-typicality, $b=-.10$, $t(3723) = 5.00$, $p<.001$, in the opposite direction as we predicted, such that feminine-stereotypic behaviors were rated as relatively more desirable for men ($M=.18$, $SD=2.38$) than for women ($M=-.07$, $SD=2.35$) and masculine-stereotypic behaviors were rated as relatively more desirable for women ($M=.06$, $SD=2.62$) than for men ($M=-.10$, $SD=2.64$) (hypothesis 3 not supported). See Fig 13 for a visualization of this interaction.

### Tests of moderations by participant identity.

Participant gender did not moderate the interaction between target gender and behavior typicality, $b<.01$, $t(3721) =.02$, $p=.981$, nor did participant race, $b=-.03$, $t(3721) =.63$, $p=.531$, or age, $b<-.01$, $t(3702) =.58$, $p=.562$. Participant political orientation did moderate the interaction, $b=-.03$, $t(3702) = -2.41$, $p=.016$, such that the interaction was stronger for more liberal participants than for more conservative participants.

**Variation in effect sizes across behaviors.** The effect sizes on the individual behavior differences in desirability for successful working men and women ranged from $d<.01$ (is sensitive and gets their feelings hurt) to $d=.47$ (takes notes in meetings); the mean effect size was $d=.19$ ($SD=.15$).

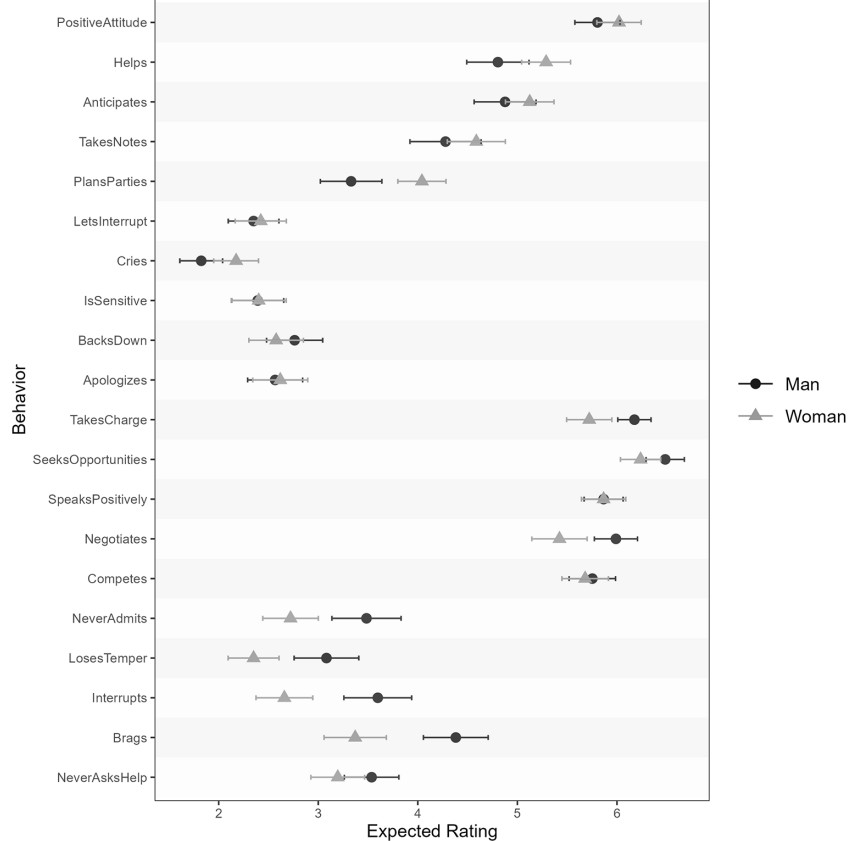

**Fig 11. Behavior-level expectedness means with 95% CIs for successful working men and women.**

**Summary.** As in Studies 1b and 2b, positive behaviors were rated as desirable in both male and female targets and negative behaviors were rated as undesirable in both male and female targets, regardless of behavior gender-typicality (hypothesis 2 supported). Unlike in Studies 1b and 2b, there was a significant two-way interaction between target gender and behavior gender-typicality, such that gender-stereotypic behaviors were seen as more desirable in the gender they are atypical of (hypothesis 3 not supported). This result is surprising and contradicts past research on stereotype desirability [5,6], which suggests that traits are more desirable in the gender they are stereotypic of, and it is unclear why this pattern occurred for successful working targets, but not for men and women in Study 1b or 2b. Overall, these results indicate that absolute behavior desirability is largely consistent for successful working men and women, although there may be small differences in the relative desirability of behaviors.

## Discussion

### Summary of results

Together, these results suggest that relative behavior expectedness differs for typical men and women, typical Black and Asian men and women, and successful working men and women, according to gender-typicality, while absolute behavior desirability is largely consistent, regardless of gender-typicality. Our results also suggest that differences in behavior expectedness are strongest for prototypical targets, and thus reactions to counter-stereotypic behaviors may be largest when researchers study gender norm violations in society, generally, compared to targets with more specific identifying

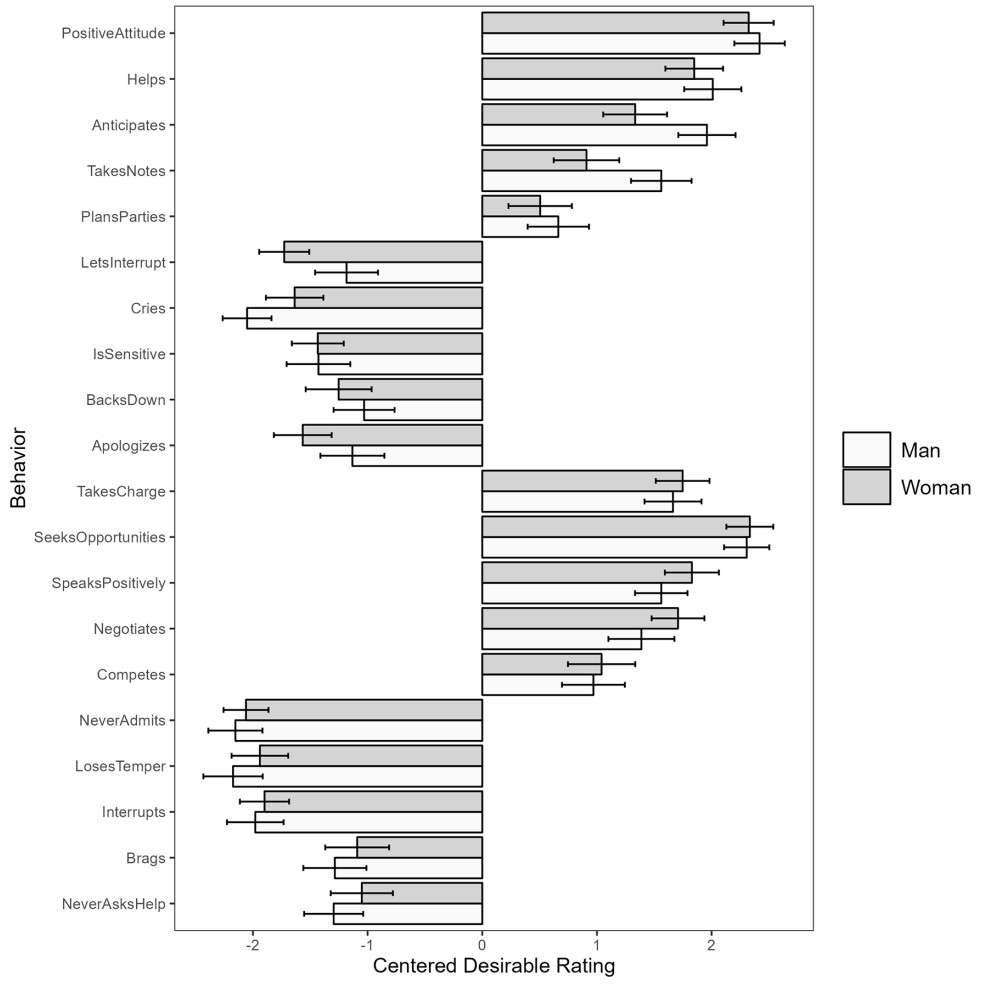

**Fig 12. Behavior-level desirability means with 95% CIs for successful working men and women.**

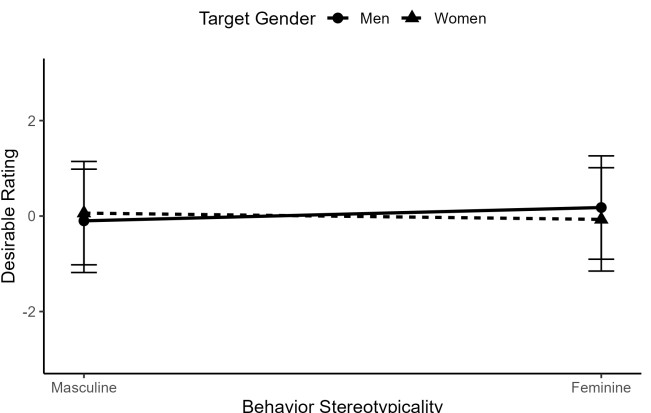

**Fig 13. The interaction between behavior gender-typicality and target gender in ratings of behavior desirability for successful working men and women.**

information, such as information about racial identity or occupation. In Studies 2a and 3a, where we investigated behavior expectedness in typical Black and Asian targets and in successful working contexts, we saw much smaller effect sizes for the differences in behavior expectedness according to gender compared to typical targets in Study 1a, suggesting that information about target race or work context may result in smaller expectancy violations for gender-inconsistent behaviors, and therefore rewards or penalties for counter-stereotypic behaviors may be weaker or less common in studies investigating reactions to Black or Asian men and women, or working men and women, compared to studies investigating reactions to prototypical men and women. This is consistent with work suggesting that White women are punished more severely for displaying dominant behaviors than Black women, as dominance is less stereotypical and therefore less expected of White women than of Black women [47].

However, we see very little support for our hypothesis that relative behavior desirability would differ between men and women. We did not see evidence that behaviors were more desirable in the gender they are stereotypic of for typical men and women, typical Black men and women, or successful working men and women – in fact, for successful working men and women, we saw evidence of the opposite effect, with behaviors rated as more desirable in the gender they are counter-stereotypic for. We only saw the predicted pattern for relative desirability in typical Asian men and women, with masculine behaviors rated as relatively more desirable in Asian men than in Asian women, and feminine behaviors rated as relatively more desirable in Asian women than in Asian men.

It is unclear why we did not see much evidence for relative desirability differences between men and women, such that behaviors are more desirable in the gender they are stereotypic of, or why we saw the opposite pattern in Study 3b for successful working men and women. There are two key differences in the methodologies of these studies that may have caused these different patterns – first, by studying behaviors, rather than traits, we may have reduced the ambiguity of the behaviors participants imagined when evaluating these desirability ratings in men and women (e.g., specifically 'taking charge in groups settings' rather than evaluating 'assertiveness,' broadly); second, by asking participants to describe their own perceptions of desirability, rather than evaluate societal perceptions of desirability, we may be tapping more closely into the actual perceptions of these behaviors, rather than perceptions of societal norms, as past work has done.

Alternatively, it is possible that society has changed in the decades since these studies were conducted, and that relative desirability differences in these behaviors are smaller than they used to be, or non-existent, or alternatively that social pressure to be egalitarian is stronger now than it used to be, resulting in a hesitance to show gender bias against gender-stereotypic men and women by giving low desirability ratings to men engaging in feminine-stereotypic behaviors, or to women engaging in masculine-stereotypic behaviors. The reversed pattern in Study 3b is particularly surprising, and suggests that there may be something unique about the working context that makes it desirable for individuals to over-come gender stereotypes and norms and act in counter-stereotypic ways; however, future work must test whether the observed differences in relative desirability generalize to more specific work settings when individuals actually enact these types of behaviors.

One caveat to our lack of support for our third hypothesis is that there was quite a bit of variability in the effect sizes of relative desirability differences across behaviors – in particular, looking at the patterns of desirability ratings for behaviors across studies, we see that the behavior "cries easily when frustrated" was consistently rated as less desirable in men than in women across all targets. Thus, there are likely to be specific behaviors that are perceived as more desirable in the gender they are stereotypic of; however, our results suggest that this is not a general pattern across all gender-stereotypic behaviors. Therefore, it may be important to investigate whether or not there are particular types of behaviors where we *do* see strong desirability differences according to gender-stereotypicality – for example, Deaux and Lewis [48] found that physical appearance was the dominant component of gender stereotyping, so it may be that behaviors related to presentation (e.g., wearing a dress vs. a suit) show the largest differences in desirability, whereas the less physical and more internal the behaviors are (e.g., backing down from a confrontation vs. interrupting someone), the smaller the differences in relative desirability.

In testing for moderations of the relative differences in expectedness by participant identity, we found numerous (but inconsistent) effects across studies. In Study 1a, we found evidence for the moderation of relative expectedness differences for male and female targets by participant gender, race, and political orientation, such that the interaction was stronger for female participants, White participants, and more liberal participants, relative to male participants, non-White participants, and more conservative participants. In Study 2a, we did not find evidence of moderating effects on this interaction, whereas in Study 3a, we found that both participant age and political orientation moderated this interaction, such that the interaction was stronger for older participants and more liberal participants, relative to more conservative participants.

We found less evidence of individual identity moderators for relative desirability differences; in Study 1b, we found evidence for a moderating effect of political orientation, such that more conservative participants seemed to show the predicted interaction (behaviors rated as more desirable in the gender they are stereotypic of), while more liberal participants did not. In Study 2b, we found evidence for moderation by age, such that the interaction was stronger for older than for younger participants, while in Study 3b, we again found evidence for a moderating effect of political orientation, such that the observed interaction was stronger for liberal participants than for conservative participants (i.e., liberal participants seem to be driving our finding that behaviors were rated as more desirable in the gender they are counter-stereotypic of, for successful working targets).

There are three possible explanations for these observed differences: first, individuals' definitions of *what* behaviors are stereotypic of men and women may differ by personal identities, second, individuals may differ in *who* they picture as a typical man or woman, and third, different individuals may differ in *how strongly* they apply gender stereotypes and norms to men and women, based on these different identities. Since these tests of moderation were only exploratory, and not consistent across target identities, we hesitate to place too much emphasis on interpreting any of these effects; however, we hope that these interactions can provide insight for researchers looking to expand on how and why individuals with varying backgrounds and identities may apply different gender-based stereotypes and norms.

Altogether, our results provide support for the implicit assumptions of expectancy violation theory and the proposed unified theory- that behavior expectedness differs between men and women, but absolute desirability is consistent, and support the application of expectancy violation theory or the proposed unified theory to predict reactions to counter-stereotypical men and women. They contradict past work suggesting that trait stereotypes broadly differ in relative desirability between men and women, and suggest that there may be situations where the unified model proposed by Eareckson and Heilman [36] can be simplified to the basic expectancy violation model, as gender may not always moderate behavior desirability.

In addition, these results point to the importance of considering individuating factors for the perceptions of gendered behaviors enacted by men and women with different identities and backgrounds, as well as in different contexts. In these studies, we examine only the individuating factor of target race, and the specific context of a general workplace. However, these are only two of many possible individuating factors and contexts for which behavioral perceptions may differ. One context where studying the differential expectations and desirabilities of gendered behaviors may be particularly important is in the domain of romantic relationships; while it has become less socially acceptable to treat men and women differently in many situations, romantic relationships (and particularly heterosexual relationships) are a domain where gender differences may still be strongly held and enforced. Our work points to the importance of considering these contextual difference, but more work must be done to broaden these findings and determine which contextual factors may result in stronger or weaker effects of gender role congruence on behavior expectedness and desirability.

## Limitations

All of our studies utilize American samples from Amazon MTurk, and it is therefore possible that these results may not generalize outside of this specific population. Another limitation of this work is that the participants in these studies are

primarily White, and the sample demographics are not nationally representative. Although not a substitute for truly representative samples, we can gain insight into the degree to which this limitation may affect our conclusions by examining whether or not participant gender and participant race (two identities that are likely to be particularly relevant for the applications of gender stereotypes and norms, based on past research) moderate the key interactions we test in each study. However, future research should specifically investigate how individuals with varying demo

In addition, the gender designation of our study targets and participants is based on a limited view of gender as a binary construct, when we know that gender identity truly exists as a continuum; our participants primarily identify as men or women, and we only collect data regarding perceptions of these behaviors in men and women. To gain a broader understanding of the patterns of behavior expectedness and desirability for men and women, it is important to replicate this work with nationally representative samples from the US and other countries, as well as examine perceptions of behavior expectedness in targets varying in identities other than race, and in contexts besides the workplace environment.

Finally, although part of our purpose in defining behaviors so simply was to define behavior gender-typicality and valence, the behaviors studied here are a sanitized version of what these behaviors would be in real men and women, both in society and in the workplace, and are only a small subset of all possible gender-stereotypic behaviors. Therefore, our findings can only truly speak to how individuals think they would feel about a general man or woman acting in the described ways; in personal interactions, with more nuanced and specific contextual information and personal histories, these perceptions will likely differ from the broad construals we study. We chose to examine high-level perceptions of gender-stereotypic behaviors by asking participants to provide their impressions of behaviors with a one-line description. In reality, these behaviors take place in complex social settings and can be enacted in a variety of ways; future researchers can build on our findings by studying perceptions of gender-typical behaviors in more specific and individual interaction contexts.

Since we were interested in testing the general assumptions of theories of reactions to gender norm violations, we aimed to develop a subset of behaviors that tapped into a variety of trait-based stereotypes, based on past work, and therefore the results we find for this subset of behaviors can provide insight into whether or not there are general patterns of differences in behavioral expectations and desirability according to target identity; however, the substantial variability in the size of the behavior-level differences in expectations and desirability reported in each study suggests that there may be certain behaviors which are more likely to be penalized, due to larger and more constant differences in relative expectedness and desirability (in these studies, crying when frustrated seemed to have the largest differences). Future work should take into account these base-level perceptions of expectedness and desirability and investigate how more complex or ambiguous situations affect perceptions of behavior expectedness and desirability in men and women who violate gender norms, as well as if there are other types of behaviors not studied here which may be particularly likely to be penalized for individuals violating gender norms.

Finally, while our results provide insight into whether the assumptions of the theories we discuss hold for perceptions of behavior expectedness and desirability in a variety of targets, they do not provide insight into whether these theories actually predict rewards and penalties for men and women engaging in gendered behaviors; future work should experimentally test whether there are differential rewards for women and men engaging in the positive, counter-stereotypic behaviors, and differential penalties for women and men engaging in the negative, counter-stereotypic behaviors studied here on both organizational outcomes, such as raises and promotions, and social outcomes, such as liking and respect.

## Conclusions

The six studies presented here lend support to the assumptions of expectancy violation theory, rather than gender congruity theories, and validate the application of the proposed unified theory of gender norm violations to study reactions to counter-stereotypic men and women. Our findings indicate that absolute behavior desirability depends on valence, not stereotypicality. Additionally, our results provide some support for the moderation of behavior desirability by gender

proposed by the unified theory of gender norm violations, specifically for typical Asian men and women. Our results also suggest that considerations of identities beyond the labels 'typical man' and 'typical woman' are critical for understanding behavior expectedness, since we see smaller effect sizes for relative differences in behavior expectedness for typical Black, typical Asian, and successful working men and women, compared to prototypical men and women; however, we see largely consistent absolute desirability ratings of gender-stereotypic behaviors across targets.

These findings are particularly intriguing when considering positive aspects of agency which current work suggests are viewed negatively and penalized in women, while being essential for organizational success – for example, taking charge in group settings, negotiating, engaging in self-promotion, seeking out leadership roles, and competing for assignments. The strong consistency of the absolute desirability of positive behaviors for all targets, regardless of the expectancy ratings, can be used to encourage individuals to engage in counter-stereotypic behaviors. There is evidence that individuals often modify or restrict their behavior out of fear of backlash for violating gender norms; our results suggest that, according to the predictions of expectancy violation theory, backlash is not inevitable when individuals engage in counter-stereotypic behaviors. These results can thus be used to assuage fears of backlash and encourage individuals to engage in positive, counter-stereotypic behaviors that can improve their quality of life or occupational and educational outcomes, such as encouraging women to negotiate for raises or seek leadership roles, and encouraging men to help others or take care of household chores. The insights from these studies can help advance both our theoretical and empirical understanding of why and when individuals may be penalized or rewarded for violating behavioral gender stereotypes, and inform the development of future studies exploring this topic.

## Author contributions

**Conceptualization:** Hannah B. Eareckson.

**Data curation:** Hannah B. Eareckson.

**Formal analysis:** Hannah B. Eareckson.

**Methodology:** Hannah B. Eareckson.

**Software:** Hannah B. Eareckson.

**Supervision:** Madeline E. Heilman.

**Visualization:** Hannah B. Eareckson.

**Writing – original draft:** Hannah B. Eareckson.

**Writing – review & editing:** Hannah B. Eareckson, Madeline E. Heilman.

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
