## [Decision Letter · Decision Letter 0]

29 Oct 2025

PONE-D-25-13132Is what’s unexpected undesirable? Investigating the expectedness and desirability of gender-stereotypic behaviors in men and womenPLOS ONE?

Dear Dr. Eareckson,

Thank you for submitting your manuscript to PLOS ONE. After careful consideration, we feel that it has merit but does not fully meet PLOS ONE’s publication criteria as it currently stands. Therefore, we invite you to submit a revised version of the manuscript that addresses the points raised during the review process.

We look forward to receiving your revised manuscript.

Kind regards,

Deni Mazrekaj

Academic Editor

PLOS ONE

Journal Requirements:

3. Please note that your Data Availability Statement is currently missing a direct link can’t access each database. If your manuscript is accepted for publication, you will be asked to provide these details on a very short timeline. We therefore suggest that you provide this information now, though we will not hold up the peer review process if you are unable.

Additional Editor Comments:

I have received reports from three expert reviewers who suggest a Major Revision. Based on my own reading of the manuscript, I agree with the reviewers and would like to invite authors to thoroughly revise their manuscript. In addition to the reviewers' comments, I also have several comments: currently, some of the key components are missing:

- What is the substantive and theoretical contribution of the study? It appears that this theoretical framework was already proposed by the authors in another paper.

- There is too little methodological information (sample and methods) to understand the results. This part should be thoroughly revised paying particular attention on whether the sample is representative and who the participants are exactly.

- The manuscript is currently too long and often repetitive. It needs to be shortened.

Reviewers' comments:

Reviewer's Responses to Questions

**Comments to the Author**

1. Is the manuscript technically sound, and do the data support the conclusions?

Reviewer #1: Yes

Reviewer #2: Yes

Reviewer #3: Yes

2. Has the statistical analysis been performed appropriately and rigorously?

Reviewer #1: Yes

Reviewer #2: Yes

Reviewer #3: No

3. Have the authors made all data underlying the findings in their manuscript fully available?

Reviewer #1: Yes

Reviewer #2: Yes

Reviewer #3: Yes

4. Is the manuscript presented in an intelligible fashion and written in standard English?

Reviewer #1: Yes

Reviewer #2: Yes

Reviewer #3: Yes

Reviewer #1: Thank you for the interesting research. Hope my comments are helpful. I have attached a referee report for this paper reviewing. The main comments are on the presentation of design of six studies and discussion on the limitations.

Reviewer #2: The study is quite interesting as it seeks to understand how gender-typicality affects how much people expect and desire certain gender-stereotypic behaviors. However, I have concerns with the demographics of the respondents especially for study 2 where there are questions asked about the expectations of blacks and asians. With a larger sample of whote population, will it be safe to draw conclusions about the traits of blacks or asians when non-blacks and non-asians are the ones being sampled. Or how do we interpret the result of the study, expectations from non-blacks or non-asians about the behaviors of blacks or asians?

Reviewer #3: Summary of the study: People often assume that when men or women act in ways that don’t match gender stereotypes, they will be judged negatively. But past research shows mixed results. This paper explores the question: are unexpected (counter-stereotypic) behaviors always seen as undesirable, or does it depend? Across six studies they ask participants to rate 20 everyday behaviors (like “taking charge in a group,” “crying when frustrated,” or “helping others”). They compare how these behaviors are judged when performed by: typical men and women, Black and Asian men and women, successful working men and women. Each behavior was rated on two things:

a) Expectedness: How much do people expect a person to act that way?

b) Desirability: Do people think that behavior is good or bad?

Participant recruitment: These are recruited from Amazon MTurk in the US.

Broad findings:

a) People expect men to act more “masculine” (assertive, competitive) and women to act more “feminine” (caring, cooperative). But these expectations vary somewhat by race and by whether the person is seen as a successful worker.

b) Desirability is mostly consistent across genders: certain traits like caring are desirable regardless of gender. Certain traits like bragging are undesirable for everyone.

c) Heterogeneity: For Asian men and women, stereotypes shaped desirability a bit more. In workplace contexts, people sometimes thought counter-stereotypic behaviors were more desirable. For example, women showing assertiveness or men being communal.

Main takeaway: Being different from gender stereotypes doesn’t necessarily hurt a person in terms of desirability. People care more about whether the behavior is positive or negative rather than whether it is gender typical.

Major comments:

1) While the authors provide demographic particulars of their participants pool, can they also use this variation to understand better who is more likely to penalize less for gender atypical behaviour? For example, are women less likely to punish women for counter gender-typical behaviour? Certain behaviour like crying seems to be clearly less desirable for men, as the authors note. Maybe pooling all studies and examining differences by men vs women participants or by participant age or electoral preferences, can throw some light on whether some demographic groups rate behaviour differently

2) Demographic composition is not similar across studies – men range from 35% to 50% - it was not clear in the paper but did the authors control for participant demographics when examining differences in expected behaviour and desirable behaviour?

3) Did the authors take any steps to minimize social desirability in reporting in survey by participants? If the authors captured any social desirability index, they should control for it in their analyses or recognize the possible role that it can play in the obtained results.

4) The statement used by authors are very generic – “How expected do you think this behavior is for a TYPICAL [MAN/WOMAN]?” on a seven-point scale ranging from ‘completely unexpected’ (0) to ‘completely expected’ (6). “ Similarly, they ask about desirability. While in general, a person can be okay with women in leadership but may not be desirable if one were to have a mother or a partner in such a role since it is not congruent with being typical of women. The authors should caveat their findings more carefully—because in a personal circumstance a trait may not be desirable even though in general it does not matter.

Minor comments:

1) The authors can consider providing what was the exact question asked in each study to the participants in the text of the paper.

**Do you want your identity to be public for this peer review?** For information about this choice, including consent withdrawal, please see our Privacy Policy

Reviewer #1: No

Reviewer #2: No

Reviewer #3: No

---

## [Author Response · Author response to Decision Letter 1]

10 Dec 2025

Review 1 Comments

The authors provide detailed information on the design and participant composition for each of the six studies in the texts. Since the studies are complementary and com-parable, I suggest creating a summary table to present this more clearly and succinctly. For example, rows could include: “Question Asked”, “Hypothesis tested”, “Number and Share of Female Participants”, “Number and Share of White Participants”, “Total Number of Participants”, and “Average Age of Participants”, with columns for each of the six studies. This would provide a more straightforward overview of sample sizes and demographics, and allow the main text to focus on differences across studies. For instance, in Study 2a and 2b, the gender imbalance is larger than in the other studies.

Thank you very much for this wonderful suggestion! We feel that this greatly improves the clarity of the designs and samples across studies, and helps shorten the body of the manuscript and reduce the repetitiveness of repeating these in each study. We have added a table with this information at the end of the joint methods section (p. 10, Table 3), although we used a different table organization than you proposed to improve the table readability.

• As acknowledged in the "Limitations" section, participants are primarily White and the samples are not nationally representative. While it is good to be transparent about this issue, I wonder whether and how this limitation might affect the results. One approach to address this would be to replicate the studies with nationally representative samples; however, obtaining such samples is not always feasible. Alternatively, some adjustments or controls might be applied to account for sample composition. For example, they “fit linear mixed models” to test hypotheses 1 and 3, which includes “random intercepts for participant ID and specific behavior”. My understanding is that they (1) use random effects for participant ID to account for repeated measures within each person as each participant rated multiple behaviors; and (2) use random effects for specific behavior to account for the differences in the average levels of expectedness or desirability across behaviors. It would be helpful if the authors could briefly discuss whether any adjustments for sample composition are feasible within this modeling framework, and or what types and directions of bias might arise.

Thank you for recommending this; we have indeed found some interesting moderations due to participant gender, race, age, and political orientation, which are now reported in the main text of the paper (pgs. 12, 13, 15, 16, 17), and summarized in the discussion (p. 19). We have compiled these moderation tests at the end of the letter and highlighted the significant moderations in each study, for ease of reference.

• A minor point: in Figures 3, 7, and 11, it is difficult to distinguish results by gender when printed in black and white. I suggest using different symbols (e.g., circles for men and triangles for women) to improve clarity.

Thank you again; we have taken your advice, and the figures have been replaced with circles and triangles to improve clarity!

Reviewer #2 Comments

The study is quite interesting as it seeks to understand how gender-typicality affects how much people expect and desire certain gender-stereotypic behaviors. However, I have concerns with the demographics of the respondents especially for study 2 where there are questions asked about the expectations of blacks and asians. With a larger sample of white population, will it be safe to draw conclusions about the traits of blacks or asians when non-blacks and non-asians are the ones being sampled. Or how do we interpret the result of the study, expectations from non-blacks or non-asians about the behaviors of blacks or asians?

Thank you for pointing this out! We have acknowledged the limitation of our sampling demographics further in the discussion, and reported tests of the moderations of participant gender, race, age, and political orientation on each of our key interactions (pgs. 12, 13, 15, 16, 17). We have compiled these moderation tests at the end of the letter and highlighted the significant moderations in each study, for ease of reference.

For the race moderation, specifically, we do observe differences in how White, Black, and Asian participants respond to the behavior expectedness in Study 2a, but we observe consistent patterns of desirability ratings across race identity in Study 2b. In Study 2a, participant race moderated the three-way interaction between participant race, participant gender, and behavior stereotypicality on expectedness, such that White participants showed gender stereotyping in expectedness for both Asian and Black targets, while Non-White participants showed gender stereotyping for Asian, but not Black, targets. Breaking this down for Asian and Black participants, we found that this effect seemed to be particularly driven by Asian participants, who showed no evidence of gender stereotyping for Black targets. We feel that including these analyses in the paper greatly enhances the interpretation of these studies, allowing for insight into how White, Asian, and Black participants responded differently (or did not respond differently) in these studies.

Reviewer 3 Comments

1) While the authors provide demographic particulars of their participants pool, can they also use this variation to understand better who is more likely to penalize less for gender atypical behaviour? For example, are women less likely to punish women for counter gender-typical behaviour? Certain behaviour like crying seems to be clearly less desirable for men, as the authors note. Maybe pooling all studies and examining differences by men vs women participants or by participant age or electoral preferences, can throw some light on whether some demographic groups rate behaviour differently

Thank you for this great suggestion for looking at the data! In line with the recommendations of Curran and Hussan’s (2009) IDA technique to test for pooled effects, we combined our data and created effects-coded variables to represent study number (in this case, variable 1 comparing Studies 1 (coded -.5) and 3 (coded .5), excluding Study 2 (coded 0), and variable 2 comparing Study 2 (coded 1) to the others (coded -.5 for both)). Since we saw numerous interactions with these study variables across our tests, and since this technique is complex, was not pre-registered in our analysis plan, and would require an additional section at the end of the paper explaining the technique and all the results (only to go back and break down these results by study, due to the significant interactions), we chose to conduct separate moderation analyses for participant gender, race, age, and political orientation; we now report these results for the main interaction (target gender * behavioral stereotypicality) across all studies in the main text. We include a summary of all moderating interactions at the end of this letter, so that you can review these findings and see if you agree with our assessment.

In addition, we try to avoid over-interpreting these results in our paper, as they were exploratory (and many), and thus may result in an inflated Type I error rate due to the increased number of tests run. We summarize them in our discussion, and conclude with this suggestion for future research: “Since these tests of moderation were only exploratory, and not consistent across target identities, we hesitate to place too much emphasis on interpreting any of these effects; however, we hope that these interactions can provide insight for researchers looking to expand on how and why individuals with varying backgrounds and identities may apply different gender-based stereotypes and norms.” (pg. 19)

2) Demographic composition is not similar across studies – men range from 35% to 50% - it was not clear in the paper but did the authors control for participant demographics when examining differences in expected behaviour and desirable behaviour?

We did not adjust for demographic differences in our key analyses, but we have added tests of the moderation of participant gender, race, age, and political orientation on each of our key analyses, which we feel address these concerns more thoroughly than including them as statistical adjustments (which would not provide insight into whether or not these evaluations differ according to demographic identity). (pgs. 12, 13, 15, 16, 17)

3) Did the authors take any steps to minimize social desirability in reporting in survey by participants? If the authors captured any social desirability index, they should control for it in their analyses or recognize the possible role that it can play in the obtained results.

Unfortunately, we did not capture a social desirability index, so we cannot adjust for that in our analyses. We have now added a paragraph explaining our choice to ask participants their personal views, rather than asking them to estimate others’ views, as research on stereotypes and norms often does (p. 6).

“Second, many investigations of gender stereotypes and norms aim to avoid social desirability effects by asking participants to evaluate how typical or desirable others in society would view a particular trait in men and women, rather than directly asking participants to provide their own evaluations of stereotypicality or desirability (i.e., Prentice and Carranza (2002) asked participants “how desirable it is in American society for a man to possess each of these characteristics”). This is very useful for assessing the perception of societal norms (i.e., how do men and women think these behaviors are viewed in society), but not as useful for assessing individuals’ own views on these behaviors.”

We acknowledge that we cannot completely eliminate this possibility, but that an anonymous online survey format should hopefully mitigate these characteristics. (p. 6)

“That is what we do in the current studies, using an online survey format to mitigate social desirability effects as much as possible. Although the motivation to appear unbiased may taint participants’ responses… asking participants their direct opinions is the best way to ascertain participants’ true views, even if it is imperfect.”

4) The statement used by authors are very generic – “How expected do you think this behavior is for a TYPICAL [MAN/WOMAN]?” on a seven-point scale ranging from ‘completely unexpected’ (0) to ‘completely expected’ (6). “ Similarly, they ask about desirability. While in general, a person can be okay with women in leadership but may not be desirable if one were to have a mother or a partner in such a role since it is not congruent with being typical of women. The authors should caveat their findings more carefully—because in a personal circumstance a trait may not be desirable even though in general it does not matter.

We very strongly agree with this statement, and we feel that our diverging results for typical, Black and Asian, and successful working women are actually in support of this assertion. Our observation that work context does change effects supports this conclusion, and future work should focus on the specific context they are studying the behavior in. We were interested here in work environments, but it is likely that specific contexts will have very different effects on perceptions of behavioral expectedness and desirability - we expand on this point in the conclusion (p. 19) when we discuss the implications of our findings for contextual differences in these evaluations, and note the domain of romantic relationships as a context of particular interest for gender differences in behavioral perceptions.

In the limitations, we also discuss further how these broad construals of behaviors may not reflect individuals’ perceptions in more individuated interpersonal contexts:

“Therefore, our findings can only truly speak to how individuals think they would feel about a general man or woman acting in the described ways; in personal interactions, with more nuanced and specific contextual information and personal histories, these perceptions will likely differ from the broad construals we study. We chose to examine high-level perceptions of gender-stereotypic behaviors by asking participants to provide their impressions of behaviors with a one-line description. In reality, these behaviors take place in complex social settings and can be enacted in a variety of ways; future researchers can build on our findings by studying perceptions of gender-typical behaviors in more specific and individual interaction contexts.” (p. 20)

Minor comments:

1) The authors can consider providing what was the exact question asked in each study to the participants in the text of the paper.

Thank you for this suggestion! We have incorporated Reviewer 1’s suggestion of a summary table including the specific question text, hypotheses, and participant demographics in each of the six studies; we believe this provides valuable clarification of the exact methods and questions asked in these studies. (p. 10)

Again, we thank you for the opportunity to revise and resubmit our paper for publication in PLOS ONE. We look forward to hearing your thoughts on this revised version of the manuscript, and believe that publishing open access in PLOS ONE is the best outlet for our work to maximize its impact and reach a large, cross-disciplinary audience.

Sincerely,

Hannah Eareckson and Madeline Heilman

Compiled identity moderation effects - significant moderations are highlighted; these are all reported in the main text of the manuscript, as well.

Study 1a: Relative expectedness differences for typical targets (included)

Participant gender moderated this interaction, b = .20, t(1878) = 6.65, p < .001, such that the interaction was significant and consistent in direction for both male and female participants, but stronger for female participants, b = .94, t(1082) = 24.85, p < .001, than for male participants, b = .55, t(740) = 11.98, p < .001. Participant race also moderated this interaction, b = .30, t(1878) = 4.54, p < .001, such that the interaction was significant and consistent in direction for both White and Non-White participants, but stronger for White participants, b = .86, t(1329) = 24.99, than for Non-White participants, b = .57, t(531) = 10.22, p < .001. Participant political orientation also moderated the interaction between target gender and behavioral stereotypicality, b = .09, t(1830) = 4.74, p < .001, such that it was stronger for liberal participants than conservative participants. Participant age was not a significant moderator, b < .01, t(1840) = .60, p = .549.

Study 1b: Relative desirability differences for typical targets (included)

Participant gender did not moderate this interaction, b = -.24, t(1878) = .84, p = .404, nor did participant race, b < -.01, t(1878) = .03, p = .974, nor age, b < .01, t(1859) = 1.76, p = .079. Participant political orientation moderated the interaction between target gender and behavior stereotypicality, b = -.04, t(1859) = 2.64, p = .008, such that more conservative participants showed the predicted interaction, rating feminine behaviors as relatively more desirable in women than in men, and masculine behaviors as relatively more desirable in men than women, but more liberal participants did not.

Study 2a: Relative expectedness differences for typical Black and Asian targets

Interaction between target gender and behavior stereotypicality (included)

Participant gender did not moderate the two-way interaction between target gender and behavior stereotypicality, b = .03, t(3793) = 1.27, p = .204, nor did participant race, b = .01, t(3793) = .21, p = .836, age, b < .01, t(3774) = 1.08, p = .282, or political orientation, b = -.01, t(3774) = .80, p = .424.

Interaction between target race and behavior stereotypicality (not included)

Participant gender did not moderate the two-way interaction between target race and behavior gender-typicality, b < -.01, t(3793) = .21, p = .837, nor did participant race, b = .07, t(3755) = 1.50, p = .134, or age, b < .01, t(3774) = .07, p = .942. The interaction was moderated by political orientation, b = .04, t(3774) = 3.18, p = .002, such that

---

## [Decision Letter · Decision Letter 1]

13 Feb 2026

Is what’s unexpected undesirable? Investigating the expectedness and desirability of gender-stereotypic behaviors in men and women

PONE-D-25-13132R1

Dear Dr. Eareckson,

We’re pleased to inform you that your manuscript has been judged scientifically suitable for publication and will be formally accepted for publication once it meets all outstanding technical requirements.

Kind regards,

Deni Mazrekaj

Academic Editor

PLOS One

Additional Editor Comments (optional):

All the comments have been addressed. Congratulations on this fine paper!

Reviewers' comments:

Reviewer's Responses to Questions

**Comments to the Author**

Reviewer #1: All comments have been addressed

Reviewer #2: All comments have been addressed

2. Is the manuscript technically sound, and do the data support the conclusions?

Reviewer #1: Yes

Reviewer #2: Yes

3. Has the statistical analysis been performed appropriately and rigorously?

Reviewer #1: Yes

Reviewer #2: Yes

4. Have the authors made all data underlying the findings in their manuscript fully available?

Reviewer #1: Yes

Reviewer #2: Yes

5. Is the manuscript presented in an intelligible fashion and written in standard English?

Reviewer #1: Yes

Reviewer #2: Yes

Reviewer #1: (No Response)

Reviewer #2: (No Response)

**Do you want your identity to be public for this peer review?** For information about this choice, including consent withdrawal, please see our Privacy Policy

Reviewer #1: No

Reviewer #2: No

---

## [Editor Report · Acceptance letter]

PONE-D-25-13132R1

PLOS One

Dear Dr. Eareckson,

I'm pleased to inform you that your manuscript has been deemed suitable for publication in PLOS One. Congratulations! Your manuscript is now being handed over to our production team.

Kind regards,

on behalf of

Prof. dr. Deni Mazrekaj

Academic Editor

PLOS One